# PBRM1 directs PBAF to pericentromeres and protects centromere integrity

Karen A. Lane [1,5], Alison Harrod [1,5], Lillian Wu [1], Theodoros I. Roumeliotis [1], Hugang Feng[1,4], Shane Foo [2], Katheryn A. G. Begg[1], Federica Schiavoni[1], Noa Amin[1], Frank T. Zenke[3], Alan A. Melcher [2], Jyoti S. Choudhary [1] & Jessica A. Downs [1]

The specialised structure of the centromere is critical for effective chromosome segregation, but its repetitive nature makes it vulnerable to rearrangements. Centromere fragility can drive tumorigenesis, but protective mechanisms preventing fragility are still not fully understood. The PBAF chromatin remodelling complex is frequently misregulated in cancer, but its role in cancer is incompletely characterized. Here, we identify PBAF as a protector of centromere and pericentromere structure with profound consequences for genome stability. A conserved feature of isogenic cell lines lacking PBRM1, a subunit of PBAF, is compromised centromere and pericentromere integrity. PBAF is present at these regions, and binding patterns of PBAF and H3K9 methylation change when PBRM1 is absent. PBRM1 loss creates a dependence on the spindle assembly checkpoint, which represents a therapeutic vulnerability. Importantly, we find that even in the absence of any perturbations, PBRM1 loss leads to centromere fragility, thus identifying a key player in centromere protection.

The centromere is a specialised region of the chromosome that is responsible for the attachment of spindle fibres via the kinetochore during mitosis, ensuring that each daughter cell receives an equal and identical set of chromosomes. Kinetochore assembly is a highly regulated process that builds upon the constitutive centromere associated network (CCAN). The CCAN bridges the CENP-A-containing centromeric chromatin with the outer kinetochore components. CENP-B contributes to this assembly by bridging CENP-A and the CCAN subunit CENP-C. Human centromeric DNA is made up of repetitive arrays of alpha satellite (α-Sat) repeats forming higher order repeats (HORs)[1]. Many α-Sat repeats contain a motif termed the B-box, which is bound by CENP-B. CENP-B is important for creating three dimensional structures in the centromere that promote appropriate chromosome segregation[2].

Centromeres are flanked by pericentromeres or transition regions, which are made up of a wider range of repetitive sequences in human cells, including α-Sat and non-α-Sat repeats, transposable elements, and duplications[1]. H3K9me3 and other repressive marks are enriched in pericentromeres and the repetitive elements are mostly silenced[3]. However, pericentromeric regions also contain non-repetitive elements, including protein coding genes, some of which are expressed[1]. Therefore, while largely heterochromatic, there are interspersed regions of open chromatin in pericentromeric regions and transition arms.

The repetitive nature of centromeric and pericentromeric sequences facilitates topological organisation, but this comes at a cost, since these repetitive sequences are vulnerable to inappropriate rearrangements[4,5]. Moreover, recent work demonstrated that cells use centromere-associated DNA breaks to help specify functional centromeres[6], which increases vulnerability if these are processed or repaired inappropriately. Pathological centromere fragility contributes to human disease, such as cancer[4]. Cells must therefore

[1]Division of Cell and Molecular Biology, The Institute of Cancer Research; London, London, UK. [2]Division of Radiotherapy and Imaging, The Institute of Cancer Research; London, London, UK. [3]Merck KGaA, Biopharma R&D, Translational Innovation Platform Oncology, Darmstadt, Germany. [4]Present address: The Francis Crick Institute; London, London, UK. [5]These authors contributed equally: Karen A. Lane, Alison Harrod. ✉e-mail: Jessica.Downs@icr.ac.uk

balance the use of these specialised and repetitive structures with mechanisms that protect genome integrity. How this is achieved is not yet fully understood.

PBRM1 (or BAF180) is a subunit of the PBAF chromatin remodelling complex, one of three mammalian SWI/SNF remodelling complexes. PBRM1 is frequently mutated in cancer, and evidence supports the idea that PBRM1 can function as a tumour suppressor[7]. Loss of function mutations are particularly prevalent in clear cell renal cell carcinoma (ccRCC), but they are also found across a range of other cancer types[8]. A critical question is what the fundamental functions of PBRM1 are, which, when lost, contribute to the development or evolution of cancer.

Here, we identify PBRM1 as a factor that prevents centromere fragility. We show that cells lacking PBRM1 have lower levels of centromere- and pericentromere-associated proteins and have altered patterns of organisation of these structures in cells. PBRM1 and the SMARCA4 subunit of PBAF are physically present at these regions, and the SMARCA4 binding pattern changes when PBRM1 is absent. Patterns of histone H3K9 methylation in centromeres and pericentromeres also change in cells lacking PBRM1. Furthermore, PBRM1 loss leads to mitotic defects and creates a dependence on the spindle assembly checkpoint, revealing a potential therapeutic vulnerability. Importantly, we find that even without external perturbations, PBRM1 loss leads to significant centromere fragility, highlighting a previously unrecognised role in centromere protection.

## Results

### Analysis of isogenic PBRM1 knockout (KO) cell lines identifies misregulation of centromere- and pericentromere-associated proteins

To identify core functions of PBRM1, we generated a panel of 17 clonal cell lines with CRISPR-Cas9 engineered loss of function mutations in PBRM1 across five different cell line backgrounds, including both cancer-derived and immortalised non-cancerous parental cell lines (Fig. 1a and ref. 9). The growth rate, cell cycle profile, and morphological changes in the knockout (KO) cells were analysed relative to the parental lines (Fig. 1b-f, and Supplementary Fig. 1). We found none of these features were substantially altered in any of the cell lines other than a modestly reduced growth rate when PBRM1 is lost (Fig. 1d and Supplementary Fig. 1).

To characterise the molecular profiles of the cells, we performed mass spectrometry on all PBRM1 KO clones and the corresponding parental lines. Pathway analysis showed alterations in chromatin organisation, DNA repair and recombination, and innate immune signalling were apparent across cell lines (Supplementary Fig. 2a and b). While apoptosis was identified as an altered pathway, we found no difference in the percentage of apoptotic cells when PBRM1 is deficient (Supplementary Fig. 2c). Previously, we found that PBRM1 is important for mediating sister chromatid cohesion at centromeres[10], raising the possibility that PBRM1 is important for chromatin structure and organisation at or near centromeres. We therefore interrogated the proteomic datasets for centromere- and pericentromere-associated proteins, including CENP-A interacting proteins, the constitutive centromere-associated network (CCAN) complex, the outer kinetochore, the chromosomal passenger complex (CPC), pericentromeric heterochromatin (PHC) proteins, and other annotated pericentromere associated proteins. Individual protein levels were modestly altered in the PBRM1 KO cells, but looking across the pathways, a trend towards lower protein levels was apparent in the PBRM1 KO cells when compared with their isogenic parental cell lines (Fig. 1g-k, and Supplementary Fig. 2d, e and 3a). In contrast, transcript levels of these genes were not consistently downregulated in the PBRM1 KO cells, indicating that the protein level changes were not primarily being driven by misregulation of gene expression (Fig. 1l, and Supplementary Fig. 3b, c). This is consistent with previous work showing that there is a poor correlation between changes in the transcriptome and the proteome of SWI/SNF-deficient cells[11], and suggests that SWI/SNF activity often influences protein stability through mechanisms other than direct regulation of gene expression. Moreover, downregulation was specific to centromere-associated complexes; by contrast, no consistent changes were apparent when centrosome proteins were analysed (Supplementary Fig. 3f, g).

We further investigated available proteomic datasets in the cancer cell line encyclopedia (CCLE) to explore whether downregulated centromere proteins are a general feature of PBRM1 loss. In the absence of isogenic comparisons, we ranked cell lines according to PBRM1 protein levels. Notably, we found a significant correlation between PBRM1 protein levels and centromere- and pericentromere-associated proteins (Supplementary Fig. 3d, e). These data suggest that lower levels of centromere- and pericentromere-associated proteins is a common feature of cells lacking PBRM1 expression.

### Loss of PBRM1 results in altered organization of centromeres and pericentromeric heterochromatin

We next set out to understand whether the decreased level of centromere- and pericentromere-associated proteins had any functional consequence on their organisation. We first looked at whether there were any detectable changes in centromere structure using FISH probes against centromere α-Sat repeats in chromosome 2 or 10. We found a modest trend towards increased area in the KO clones compared with the parental RPE1 or 1BR3 cells (Fig. 2a–c and Supplementary Fig. 4). This pattern was previously observed in CENP-A-depleted cells[12] raising the possibility that PBRM1 loss leads to CENP-A deficiency. Since CENP-A was not present in our proteomic dataset, we interrogated CENP-A by immunofluorescence (IF). However, when CENP-A signal area and shape (eccentricity) were measured, we found no clear difference in the PBRM1 KO cells (Supplementary Fig. 5a-c), suggesting that CENP-A deficiency is not driving the changes in α-Sat signal in the PBRM1 KO cells. Since α-Sat repeats are also present outside of CENP-A-containing chromatin, one possible explanation for the slight changes in FISH signals is a failure of these regions to form appropriate folded structures in the KO cells, leading to an increased volume of α-Sat repeat-containing chromatin in the KO cell nuclei.

We used IF to interrogate CENP-B and NDC80 patterns to further understand the impact of PBRM1 loss on centromere-associated structures. While there was no difference in the CENP-B signal area, we found a modest, but reproducible decrease in eccentricity scores in the PBRM1 KO cells (Supplementary Fig. 5d–f), suggesting that the three-dimensional organisation of the region is impacted by the loss of PBRM1. NDC80 patterns were also subtly altered in PBRM1 knockout cells, with an increase in both area and eccentricity (Supplementary Fig. 5g–i). Interestingly, we also noticed an increase in the distance between mitotic sister chromatid-associated NDC80 signals in the PBRM1 KO cells (Supplementary Fig. 5j), consistent with a change in the three-dimensional organisation of pericentromeric regions.

We next examined whether there were any microscopically detectable changes in pericentromeric heterochromatin. We performed immunofluorescence using an antibody against H3K9me3, a marker of heterochromatin, and found that the intensity of this signal is subtly but consistently increased in the absence of PBRM1 (Fig. 2d, e), suggesting altered organisation or prevalence of heterochromatin in these cells. To see whether there were changes specifically in pericentromeric heterochromatin, we used an antibody against CENP-A to identify centromeres and then quantified the surrounding H3K9me3 signal (Fig. 2f). Interestingly, we find that the pattern of H3K9me3 intensity changes in the PBRM1 KO clones (Fig. 2g). The signal was lower in the shell closest to CENP-A, but higher in outer shells in the PBRM1 KO cells, again suggesting that PBRM1 helps to organise chromatin in regions surrounding centromeres.

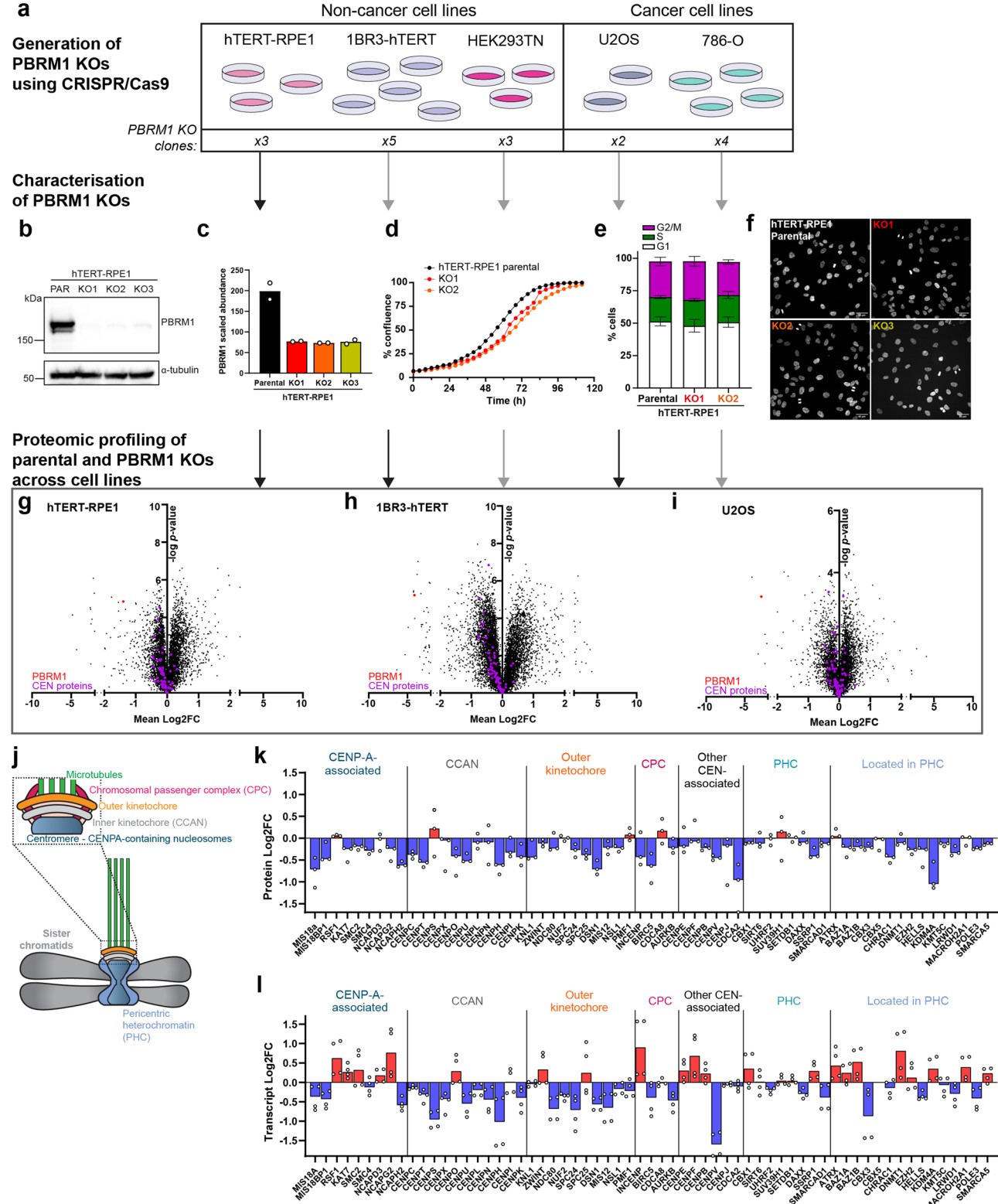

## PBRM1 and SMARCA4 are present at pericentromeres, and PBRM1 loss leads to changes in SMARCA4 association

One possible mechanism by which PBRM1 is regulating the structure and organisation of chromatin and associated proteins in centromeres and pericentromeric regions is through direct remodelling activity in these regions. There is some evidence to support this possibility. PBAF was reported to associate with kinetochores of mitotic chromosomes[13], and SWI/SNF subunits have been identified in protein-interaction studies of centromere-associated proteins, including CENP-C, INCENP, and Bub1[14–16]. However, in contrast to our understanding of PBAF binding patterns in euchromatic regions of the genome, little is known about the specific binding pattern of PBAF in repetitive regions.

We therefore set out to determine whether PBAF associates with centromeric or pericentromeric chromatin (including HORs, Hsat (Human satellite) repeats, transition (ct) arms; Fig. 3a) and gain a

**Fig. 1 | Analysis of isogenic PBRM1 knockout (KO) cell lines identifies centromere associated protein misregulation as a common feature. a** Workflow for generation of PBRM1 knockouts in a panel of cell lines. Number of independent clones validated for each cell line is indicated at the bottom. **b**–**f** Characterisation of PBRM1 knockouts using the hTERT-RPE1 cell line as an example. **b** Western blotting of whole cell lysates from parental and PBRM1 KO cells for PBRM1. α-tubulin is used as a loading control. **c** Scaled abundances of PBRM1 in proteomic analyses of whole cell protein extracts. Points correspond to independent biological replicates ($n = 2$). **d** Proliferation of RPE1 parental and two PBRM1 KO clones, measured using phase contrast Incucyte images ($n = 2$). **e** Cell cycle distribution of RPE1 parental and two PBRM1 KO clones measured using flow cytometry. $n = 4$, mean ± SEM, data were non-significant (ns) based on a 2way ANOVA using Dunnett's multiple comparisons test. **f** Immunofluorescence images of nuclear morphology in RPE1 parental and PBRM1 KO clones. Scale bar corresponds to 40 μm. **g**–**i** Protein abundances in PBRM1 knockouts compared to parental cells in hTERT-RPE1 (**g**), 1BR3-hTERT (**h**),

and U2OS (**i**) cell lines, detected using LC-MS of whole cell protein extracts. The mean log2 fold change (Log2FC) of protein abundance in PBRM1 knockouts versus parental cells is plotted against the -log $p$ value, calculated using two-sided one-sample t-test. PBRM1 is highlighted in red, while centromere- & pericentromere-associated proteins are highlighted in purple. **j** Schematic outlining regions of the centromere and pericentric heterochromatin, including the kinetochore in mitosis. **k** Median Log2FC of annotated centromere- & pericentromere-associated proteins in RPE1 PBRM1 knockouts compared to parental cells. **l** Transcript levels of annotated centromere- & pericentromere-associated genes corresponding to the proteins in k detected using RNA-seq. Median Log2FC of annotated genes transcribing centromere- & pericentromere-associated proteins in RPE1 PBRM1 knockouts was plotted compared to parental cells. Points in **k** and **l** correspond to individual knockout clones from two independent biological replicates. Source data are provided as a source data file.

comprehensive view of PBAF localisation patterns. To do this, we performed CUT&RUN sequencing in both low and high salt conditions to ensure the capture of heterochromatic regions[17]. We mapped PBRM1 and SMARCA4 (BRG1), one of two catalytic subunits of PBAF, using both IgG and a SMARCA4 KO cell line (Fig. 3a and Supplementary Figs. 6 and 7b) as negative controls, and CENP-B as a positive control. Because of the repetitive nature of the region, we analysed the data in two ways: peak calling of uniquely mapping reads, and a *k*-mer analysis of reads mapping to these regions, thus allowing analysis of reads mapping to multiple locations (Fig. 3a and Supplementary Fig. 6, described below).

We first aligned uniquely mapping reads using the reported telomere-to-telomere (T2T) CHM13 genome[1,18]. CENP-B localised primarily to HORs, as expected, with some additional sites of enrichment in flanking regions (Fig. 3b and Supplementary Fig. 7a). We found enrichment of SMARCA4 and PBRM1 primarily in pericentromere sequences when compared with negative controls, and most SMARCA4 and PBRM1 peaks were located in the transition (ct) arms (Fig. 3b and Supplementary Fig. 7a). As expected, many of the PBRM1 and SMARCA4 peaks overlapped, but non-overlapping peaks were also identified (Fig. 3c, Supplementary Fig. 7c, e and f). This likely reflects the combinatorial flexibility of the complexes (i.e. PBAF can contain SMARCA2 instead of SMARCA4, and SMARCA4 is found in other SWI/SNF complexes).

When analysing the peaks found in centromeres and pericentromeric regions, we find that the peak distribution profile of PBRM1 has a stronger association with promoters when compared with SMARCA4 (Fig. 3d). This profile is similar to that of the genome-wide distributions of both PBRM1 and SMARCA4 in our dataset (Supplementary Fig. 7d), and is consistent with previous studies showing that, in euchromatin, PBRM1-containing PBAF complexes are enriched at both promoters and enhancers, whereas BAF complexes are more often found at enhancers[19,20]. To determine whether pericentromere-association is a conserved feature of PBAF, we interrogated datasets in which PBRM1 and SMARCA4 were mapped[21,22], and strikingly, we find that the association of PBRM1 and SMARCA4 in these regions is apparent in other cell line models (Fig. 3e).

Because of the repetitive nature of these regions, we also analysed centromere- and pericentromere-associated reads that mapped to multiple locations, and were therefore unable to be precisely mapped, by performing a *k*-mer analysis[1,23]. This analysis identifies 51-mer sequences that are significantly enriched in the mapping datasets (see Fig. 3a and Supplementary Fig. 6 for workflow) relative to the IgG control. CENP-B was analysed relative to IgG as a positive control.

In order to look for binding patterns and proximity to other features, enriched centromere- and pericentromere-associated *k*-mers were mapped back onto the T2T genome. The CENP-B-associated *k*-mers map primarily to the active higher order repeats (HORs) where CENP-B is known to bind (Fig. 3f, i, and Supplementary Fig. 7g, h), and

motif analysis of the CENP-B associated *k*-mers identified the CENP-B box (Fig. 3h), providing support for the utility of this approach.

Similar to peak enrichment, we found that the majority of PBRM1- and SMARCA4-associated *k*-mers are located within the transition arms, but a small proportion map to repeats (Fig. 3f, i Supplementary Fig. 7g, h), which may indicate some association across these regions. Again, we find both overlapping as well as PBRM1- and SMARCA4-specific *k*-mers (Fig. 3g and Supplementary Fig. 7i).

We additionally mapped SMARCA4 in PBRM1 KO cells to interrogate changes to PBAF binding patterns when PBRM1 is absent. A subset of SMARCA4-enriched peaks and SMARCA4-enriched *k*-mers are lost when PBRM1 is absent (Fig. 4a and Supplementary Fig. 8a), suggesting that PBRM1 is important for targeting SMARCA4 and PBAF to these locations. Interestingly, a considerable number of SMARCA4-enriched peaks and SMARCA4-enriched *k*-mers are gained when PBRM1 is deficient (Fig. 4a, b and Supplementary Figs. 8 and 9a), suggesting that PBRM1 loss leads to aberrant SMARCA4 binding at sites not normally bound by SMARCA4. Together, these data indicate that PBAF binds to specific sites in chromatin flanking centromeres, and this binding pattern is disrupted when PBRM1 is deficient.

## Distinct patterns in PBRM1-dependent and -independent SMARCA4 association with pericentromeric regions

To further explore binding patterns, we divided centromere- and pericentromere-specific sites of PBRM1 and SMARCA4 chromatin association (both peaks and *k*-mers) into three different groups (Fig. 4c, d, and Supplementary Fig. 9b, c). First, we looked at all PBRM1 enriched sites that did not have SMARCA4 enrichment in the PBRM1 KO cells (PBRM1 specific), representing sites that are dependent on PBRM1. Second, we grouped PBRM1 enriched sites present at locations where SMARCA4 is still bound when PBRM1 is deficient (PBRM1 non-specific), which likely represents other SWI/SNF complexes, such as BAF. Third, we grouped all SMARCA4 enriched sites that appear only when PBRM1 is deficient (KO specific), which reflect potentially aberrant or inappropriate binding when PBRM1 is deficient). As expected, a greater proportion of PBRM1-specific sites are located in promoter regions when compared with PBRM1 non-specific sites (Fig. 4e). The majority of enriched *k*-mers from all three groups were located in ct arms, suggesting there is no substantive shift into other centromeric regions (Supplementary Fig. 9d, e).

We then compared the location of these groups of enriched sites (peaks and *k*-mers) to mapped PTMs. Again, consistent with genome-wide binding patterns of BAF and PBAF, we find that the PBRM1 specific peaks are associated with both promoter- and enhancer-associated PTMs, such as H3K4me3, and PBRM1 non-specific SMARCA4 peaks show relatively more association with enhancer-associated PTMs, such as H3K27ac (Supplementary Fig. 10a). Interestingly, the SMARCA4 peaks that arise when PBRM1 is deficient (KO specific) display a distinct pattern from either of the other two groups.

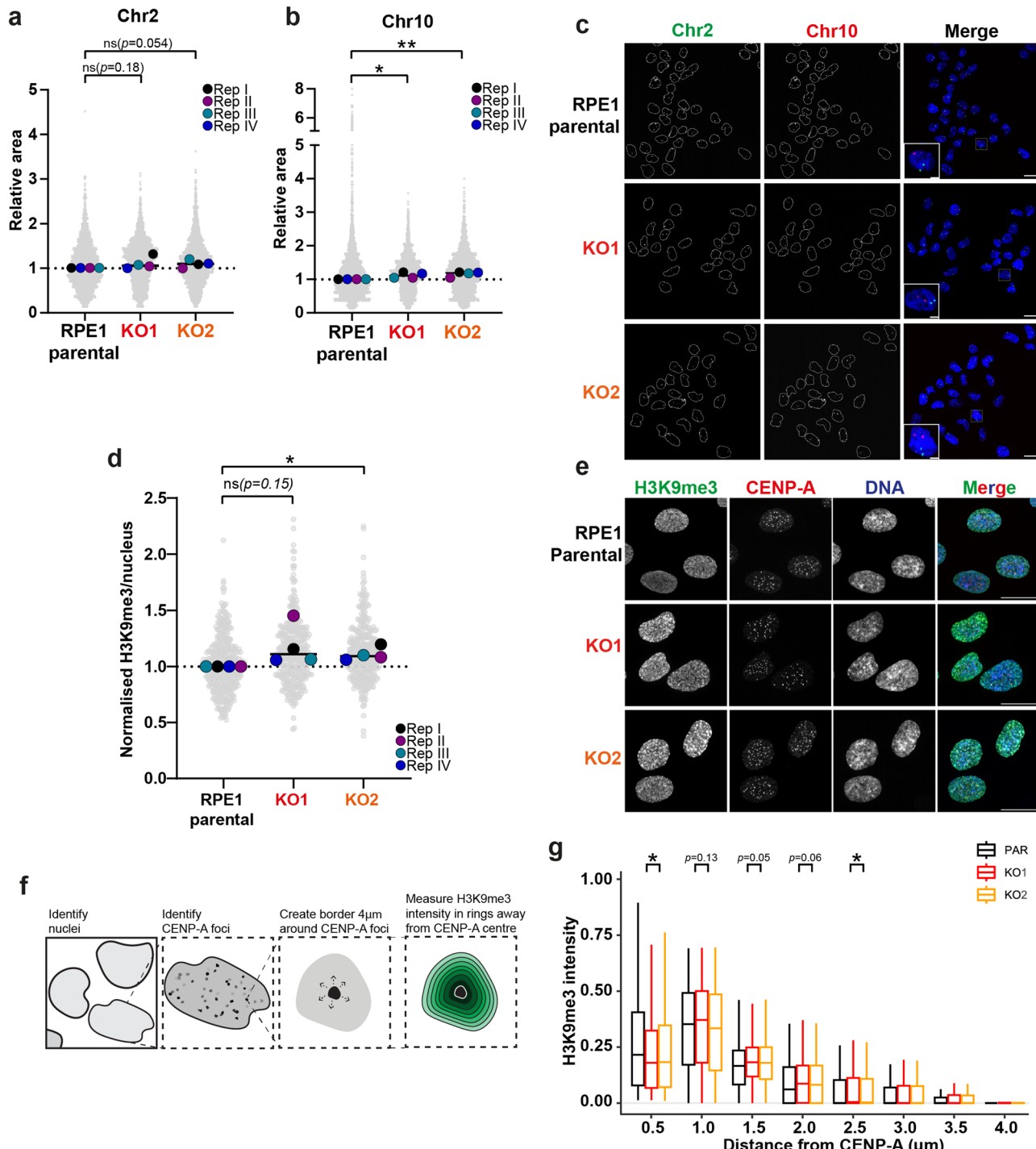

**Fig. 2 | PBRM1 KO cells display increased H3K9me3 intensity around centromeres. a, b** Quantification of the area of individual foci in cells stained for α-satellite centromeric regions in (**a**) chromosome 2 and (**b**) chromosome 10, using FISH probes. $n = 3$, coloured points represent median of biological replicates, line = median of 3 replicates, data were normalised to median area of parental cells and analysed using two-sided t-test of experiment medians, $*p = 0.0338$ $**p = 0.0061$. **c** Representative images of α-satellite FISH of chromosomes 2 and 10 in RPE1 parental and PBRM1 knockouts. Scale bars correspond to 20 μm; or 5 μm in inset images. **d** Quantification of H3K9me3 signal per nucleus normalised to median intensity of parental nuclei. Grey points correspond to individual nuclei. $n = 4$, coloured points represent median of biological replicates, line = median of 4 replicates. At least 325 nuclei were analysed per condition and data were analysed using two-sided t-test of experiment medians, $*p = 0.0360$. **e** Representative images showing H3K9me3 and CENP-A signal in RPE1 parental and PBRM1 KO cells. Scale

bars correspond to 20μm. **f** Schematic describing the method of quantifying H3K9me3 signal around centromeres, which were defined by the presence of CENP-A. Briefly, a Cell Profiler pipeline was used to identify nuclei and CENP-A foci within each nucleus. CENP-A signal was expanded to a total distance of 4 μm from the centre of each CENP-A focus, and divided into 8 rings. H3K9me3 intensity was measured in each ring. **g** Boxplot indicating H3K9me3 intensity at increasing distances from the centre of CENP-A foci in RPE1 parental and PBRM1 KOs. Boxes contain the 25th to 75th percentiles with a line at median, and whiskers extend to the largest value no further than 1.5 times the inter-quartile range, $n = 3$. CENP-A foci in at least 290 nuclei quantified per condition and data were analysed using two-sided t-test of experiment medians corrected for multiple comparisons using the Holm-Sidak method (threshold = 0.05), $*p = 0.0485$ and $*p = 0.0272$ for 0.5 μm and 2.5 μm distances, respectively. Source data are provided as a source data file.

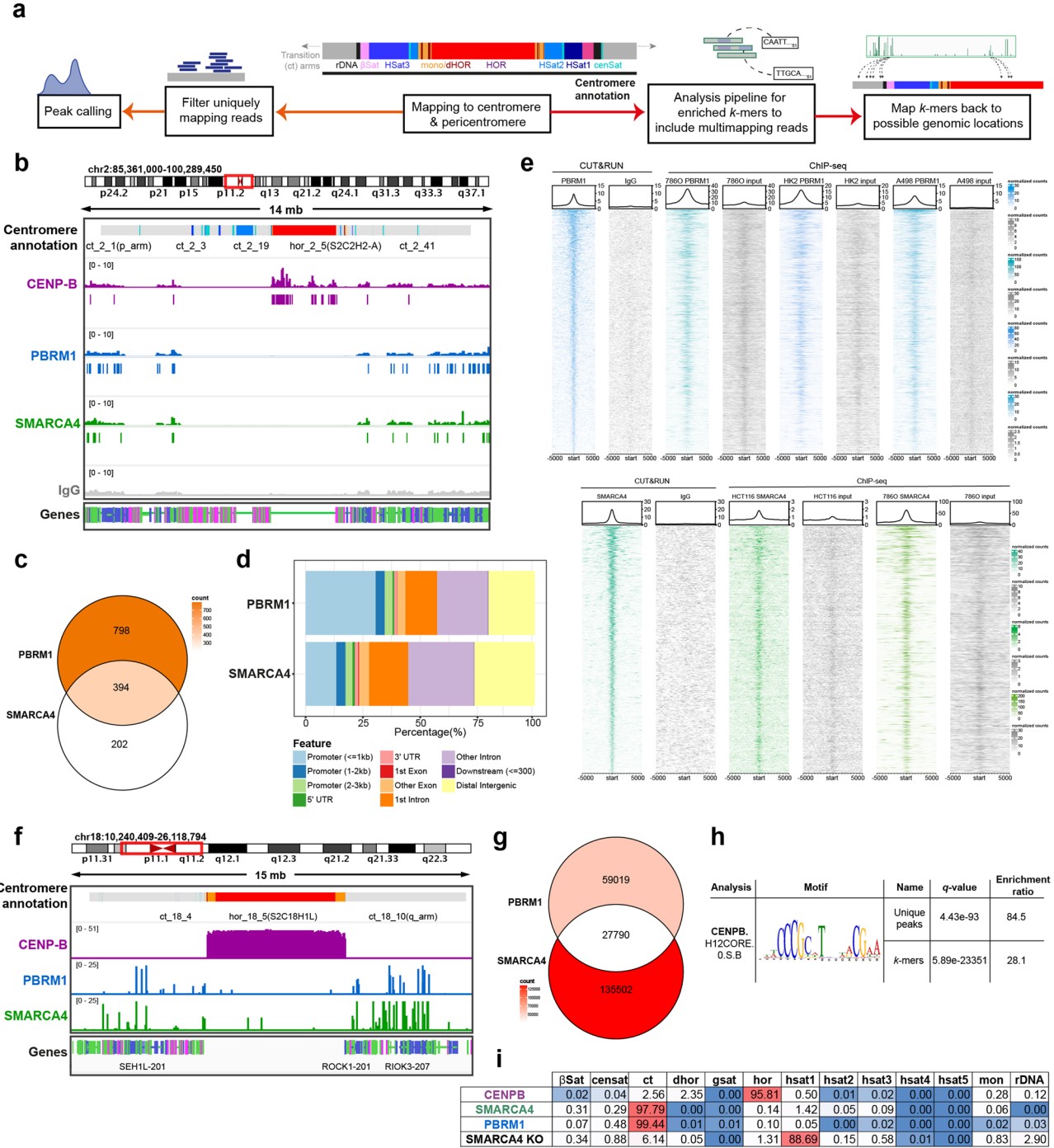

**Fig. 3 | SMARCA4 is present at centromeres, and this pattern changes in PBRM1 KOs. a** Simplified flowchart describing the mapping strategy to centromeric and pericentromeric sequences (detailed version in Supplementary Fig. 6). **b** Representative genome tracks displaying coverage of reads from CENP-B (purple), PBRM1 (blue), SMARCA4 (green) and IgG control (grey) CUT&RUN sequencing in RPE1 parental cells across the centromere. **c** Venn diagram indicating the overlap of significantly enriched SMARCA4 and PBRM1 peaks in centromeric and pericentromeric regions in RPE1 parental cells. **d** Stacked colour bar representing the genomic distribution of enriched PBRM1 and SMARCA4 peaks, categorised by feature, within centromeric and pericentromeric regions. **e** CUT&RUN and ChIP-seq signal heatmaps (PBRM1 – top, blue; SMARCA4 – bottom, green) in the indicated cell lines +/−5kb from the centre of RPE1 PBRM1 or SMARCA4 CUT&RUN peaks in the centromere and pericentromere, versus the IgG

or input control (grey) of each experiment. Peaks are ordered by signal of the leftmost heatmap, i.e. CUT&RUN peaks. An average signal plot is shown at the top of each heatmap. **f** Representative genome tracks displaying mapping locations of enriched *k*-mers across the centromere and pericentromere from analysis of CENP-B (purple), PBRM1 (blue), and SMARCA4 (green) CUT&RUN sequencing in RPE1 parental cells. **g** Venn diagram indicating the overlap of enriched *k*-mers (FC > 2) in centromeric and pericentromeric regions, in SMARCA4 and PBRM1 in RPE1 parental cells. **h** CENP-B motif detection following motif analysis of CENP-B-enriched *k*-mers compared to a shuffled control, where CENP-B *k*-mer sequences were shuffled maintaining 3-mer frequencies. **i** Percentage of enriched *k*-mers that map to specific regions in the centromere and pericentromere in each dataset, including SMARCA4 *k*-mers enriched in SMARCA4 KO cells (negative control).

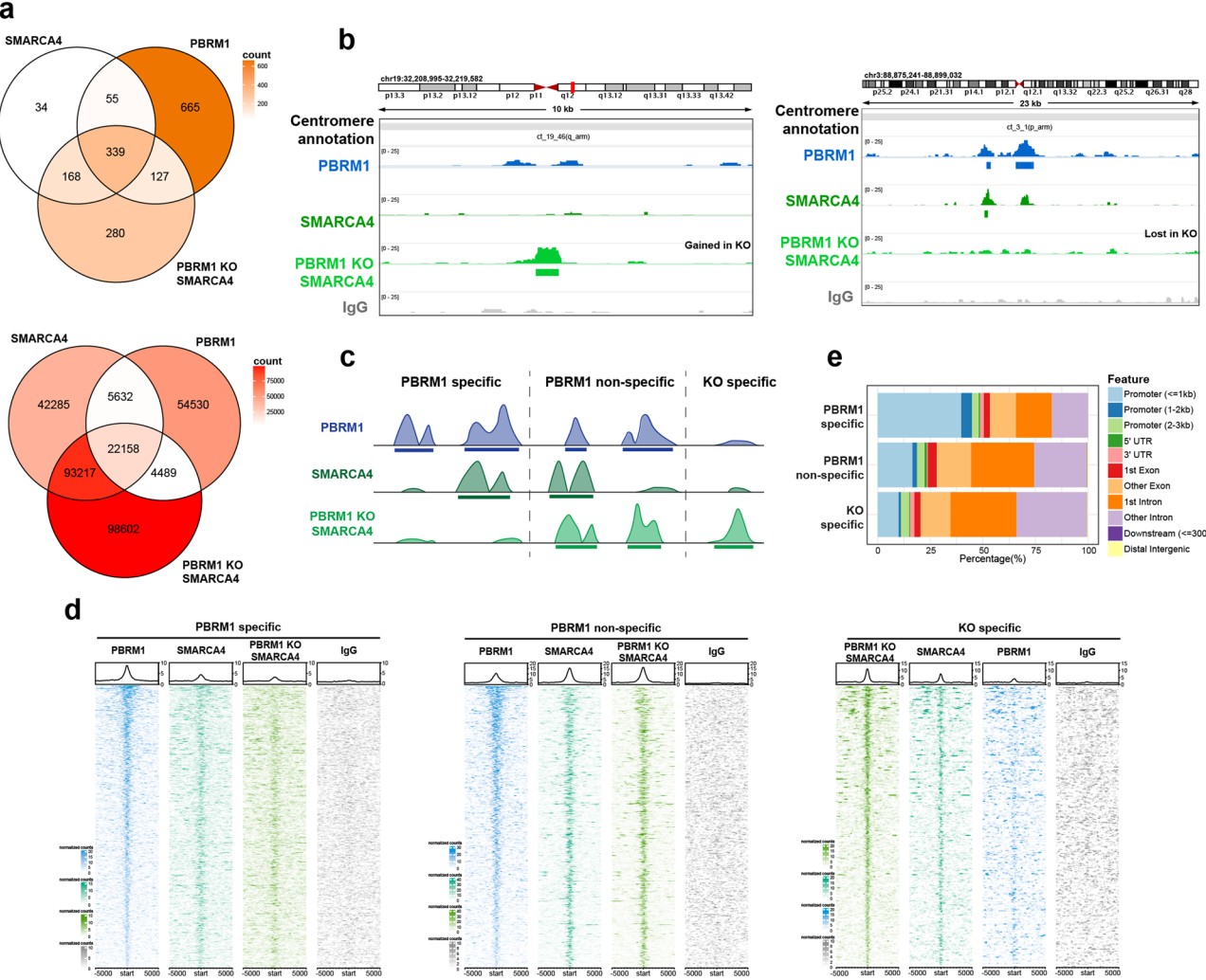

**Fig. 4 | SWI/SNF enrichment at a subset of centromeric chromatin marks is altered in a PBRM1-dependent manner. a** Venn diagram indicating the overlap of significantly enriched peaks (top, orange) or *k*-mers (bottom, red) in the centromere and pericentromere, of SMARCA4 and PBRM1 in RPE1 parental cells, and SMARCA4 in PBRM1 knockout (KO) cells. Venn diagrams show enriched peaks or *k*-mers found in at least two of three independent biological replicates, versus their IgG controls. The colour corresponds to the total number of enriched peaks or *k*-mers in each region of the Venn diagram (count). **b** Representative genome tracks displaying coverage of reads from PBRM1 (blue), SMARCA4 (green), and IgG control (grey) CUT&RUN sequencing in RPE1 parental cells and SMARCA4 in PBRM1 knockout cells (light green), showing an example of peaks gained (left) or lost (right) in the PBRM1 knockout cells. One representative independent biological replicate is shown, with boxes underneath representing peaks that were called as significantly enriched ($q < 0.01$) in at least two of three replicates versus their IgG control. Genomic location is indicated at the top and centromeric annotation is shown above the tracks. **c** Schematic showing PBRM1 specific, PBRM1 non-specific and KO specific categories of peaks from CUT&RUN sequencing. Example graphics for called peaks are shown for each category. **d** CUT&RUN signal heatmaps representing the three categories of peaks in the centromere and pericentromere. Signal +/− 5 kb from the centre of each peak is shown for PBRM1 (blue) and SMARCA4 in parental RPE1 (green), SMARCA4 in PBRM1 knockout cells (light green), and IgG control (grey). Peaks are ordered by signal of the left-most heatmap for each condition. An average signal plot is shown at the top of each heatmap. **e** Stacked colour bar representing the genomic distribution of the three categories of peaks, categorised by feature, within centromere and pericentromere regions.

There is a decrease in association with regions normally enriched in H3K4me3, H3K9ac, and H3K36me1, and an increase in association with H3K36me3-enriched chromatin (Supplementary Fig. 10a). In addition, when enriched *k*-mers were analysed, we noticed a small but potentially important PBRM1-dependent change in association with H3K9me3 (Supplementary Fig. 10a, b). The number of KO-specific *k*-mers that intersect with H3K9ac enriched *k*-mers is lower than for the PBRM1 specific subset, similar to what is seen with H3K4me3, and instead, there is a greater number of KO-specific *k*-mers that intersect with H3K9me3 enriched *k*-mers. (Supplementary Fig. 10b). To look at this more quantitatively, we analysed the proximity of these by plotting the relative distance between the mapping locations of subsets of SWI/SNF *k*-mers and histone PTM *k*-mers as a cumulative fraction[24]. We used the promoter-associated H3K4me3 mark as a positive control, which showed, as expected, that PBRM1-specific mapping locations are closer to H3K4me3 than PBRM1 non-specific *k*-mers (Supplementary Fig. 10c). Interestingly, KO specific *k*-mer mapping locations tend to be further away from those of H3K4me3 *k*-mers. In contrast, these are much closer to mapping locations that are normally enriched in H3K9me3 and further away from those enriched in H3K9ac (Supplementary Fig. 10c, d).

## Changes in centromere- and pericentromere-associated H3K9 methylation patterns are apparent in the absence of PBRM1

Given the changes evident in H3K9me3 staining (Fig. 2) combined with differences in mapping locations relative to H3K9 marks (Supplementary Fig. 10), we tested the possibility that loss of PBRM1 would lead to altered H3K9 methylation patterns. To do this, we performed

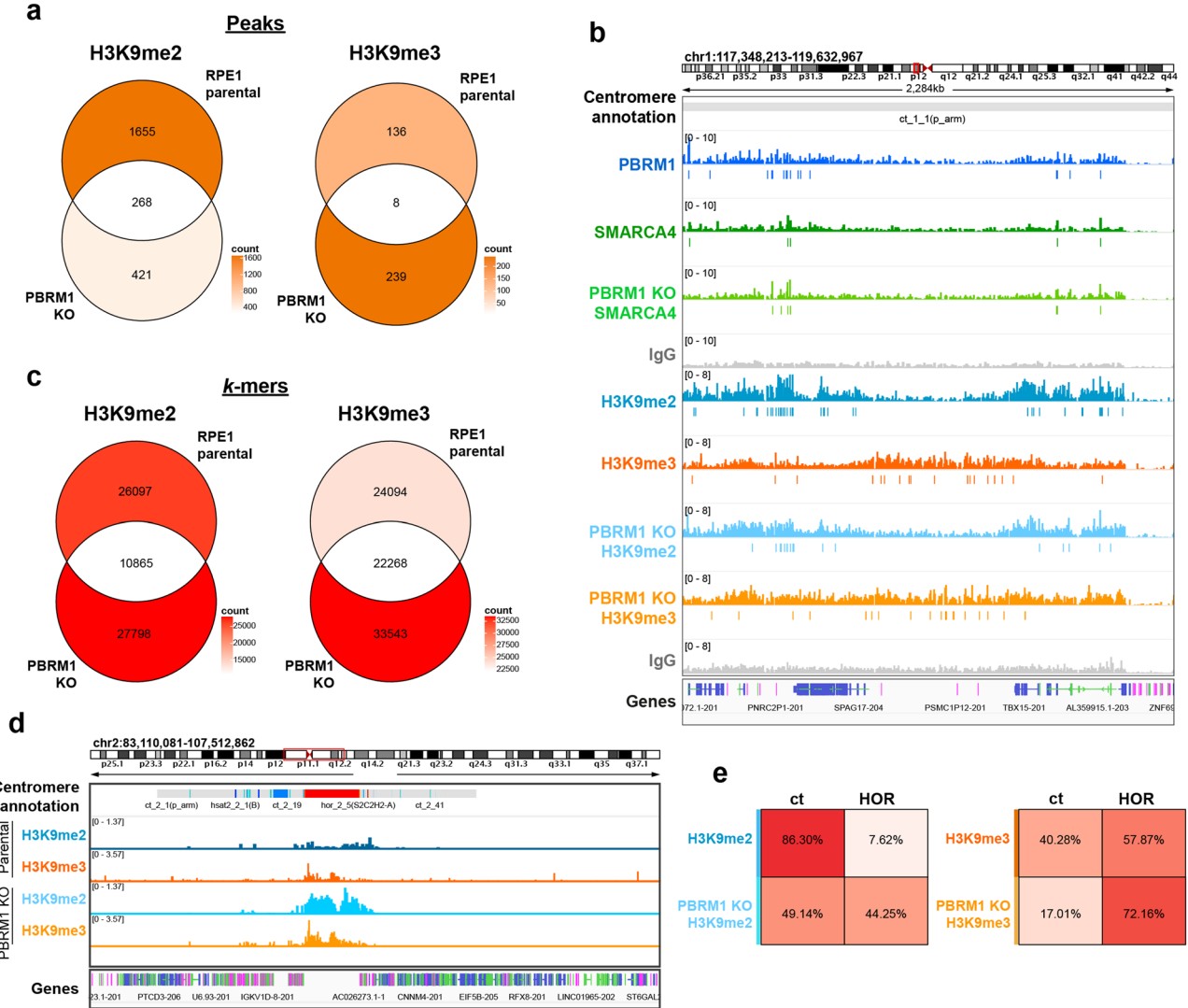

**Fig. 5 | H3K9 methylation patterns in the centromere and pericentromeric regions are altered in the absence of PBRM1. a** Venn diagram indicating the overlap of significantly enriched H3K9me2 and H3K9me3 peaks in the centromere and pericentromere in RPE1 parental and PBRM1 knockout (KO) cells.
**b** Representative genome tracks displaying coverage of reads from PBRM1 (blue), SMARCA4 (green), H3K9me2 (teal), H3K9me3 (orange) and IgG (grey) in RPE1 parental cells, and SMARCA4 (light green), H3K9me2 (light blue) and H3K9me3 (light orange) in PBRM1 KO cells. One representative independent biological replicate is shown, with boxes underneath representing peaks that were called as significantly enriched (q < 0.01) in at least two replicates versus their IgG control. **c** Venn diagram indicating the overlap of significantly enriched H3K9me2 and H3K9me3 k-mers in the centromere and pericentromere in RPE1 parental cells and PBRM1 knockout (KO) cells, versus the average IgG control. **d** Representative

genome tracks displaying mapping locations of enriched k-mers across the centromere and pericentromere from analysis of H3K9me2 (teal) and H3K9me3 (orange) in RPE1 parental cells and PBRM1 KO cells (light blue and light orange, respectively). **e** Percentage of enriched k-mers that map to the transition arm (ct) or the HOR in parental and PBRM1 KO RPE1 cells, plotted as a heatmap, for H3K9me2 (left) and H3K9me3 (right). The colour scale of white to red represents low to high percentages, with specific percentages indicated. Venn diagrams show enriched peaks (**a**, orange) or k-mers (**c**, red) found in at least two out of the three independent biological replicates, versus their IgG controls. The colour corresponds to the total number of enriched peaks or k-mers in each region of the Venn diagram (count). For genome browser tracks (**b,d**), genomic location is indicated at the top, centromeric annotation is shown above the tracks and transcript annotation (Genes) is below.

CUT&RUN mapping of H3K9me2 and H3K9me3 in parental and PBRM1 KO RPE1 cells.

When uniquely mapping reads were mapped to the centromere and pericentromeric regions, we found a shift in the distribution of both marks. Overall, there was a loss of H3K9me2 peaks and a gain of H3K9me3 peaks in the PBRM1 KO cells (Fig. 5a), and the H3K9me3-signal appears to spread into regions formerly occupied by H3K9me2 (Fig. 5b and Supplementary Fig. 11b). Redistribution of these marks was also apparent genome-wide (Supplementary Fig. 11a).

Again, we analysed enriched repetitive sequences using k-mer analysis. In doing so, we found a shift in the distribution of H3K9

methylation marks associated with repetitive DNA when PBRM1 is deficient (Fig. 5c), with both losses and gains of k-mers apparent in the PBRM1 KO cells. Interestingly, we found that both H3K9me2- and H3K9me3-enriched k-mers were lost from ct arms and gained in HOR sequences when PBRM1 is deficient (Fig. 5d, e, Supplementary Fig. 11c–e). A loss of H3K9me3-enriched k-mers from divergent HORs and monomeric repeats is also apparent, while H3K9me3 association with Hsat1 sequences increases when PBRM1 is deficient (Supplementary Fig. 11c–h). Together, these data demonstrate that loss of PBRM1 alters the pattern of H3K9 methylation in centromeres and flanking pericentromeric sequences.

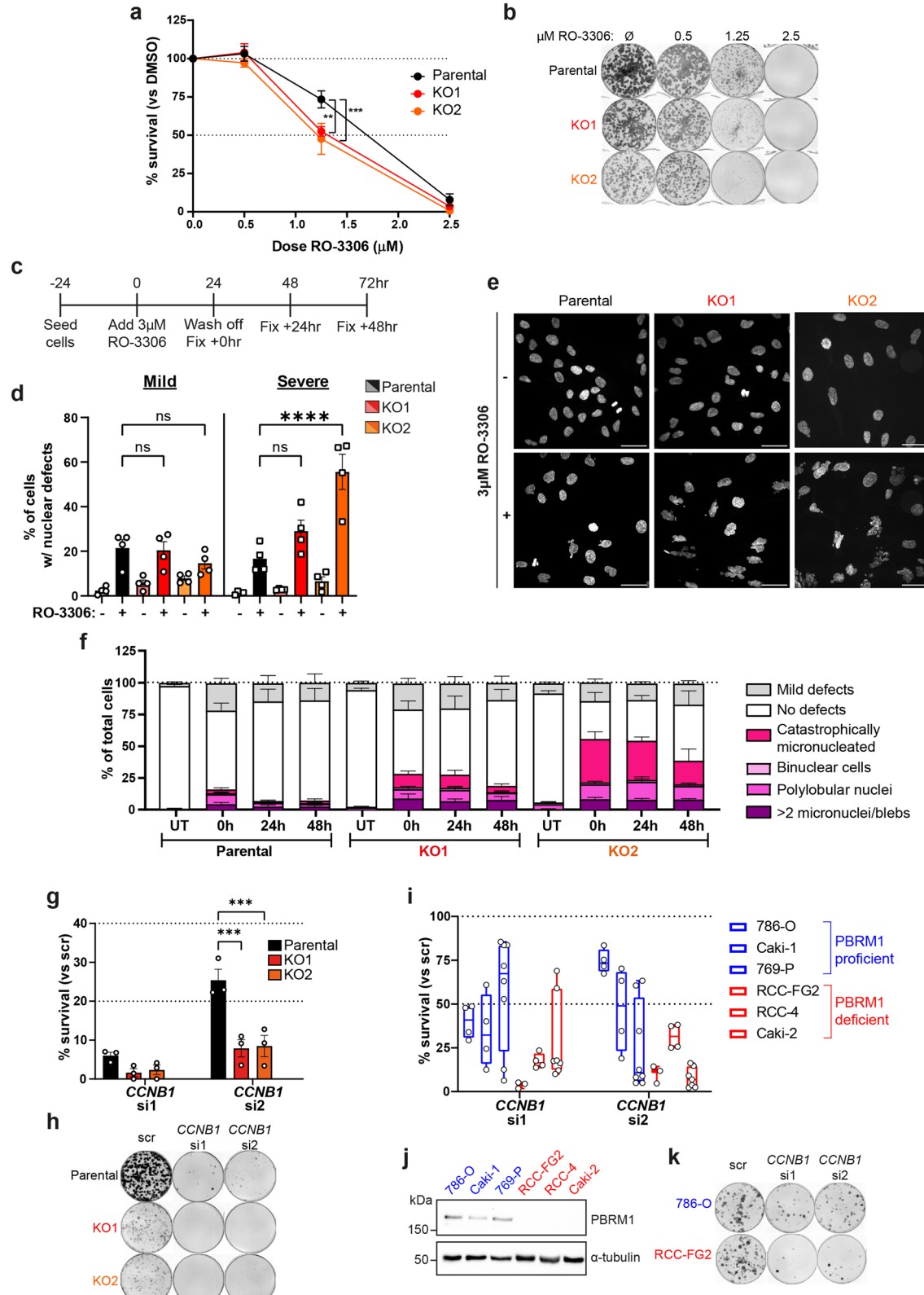

### PBRM1 KO cells are sensitive to mitotic perturbation

We next wanted to test whether the changes in structure at centromeres and pericentromeric regions where PBRM1 is bound lead to sensitivity to mitotic perturbation when PBRM1 is absent. We tested this and found PBRM1 KO cells are sensitive to chronic CDK1 inhibition compared with parental RPE1 cells (Fig. 6a, b). We also treated cells with an acute dose of CDK1 inhibitor (Fig. 6c) and monitored the

presence of aberrant mitotic events after release. PBRM1 KO cells show a substantial increase in the number of nuclear abnormalities (Fig. 6d–f, Supplementary Fig. 12a, b).

We also found PBRM1 KO cells are selectively sensitive to depletion of CCNB1 (Fig. 6g, h and Supplementary Fig. 12c). 786-O derived PBRM1 KO clones, however, which do not show downregulation of centromere- or pericentromere-associated proteins (Supplementary

**Fig. 6 | PBRM1 KO cells are sensitive to mitotic perturbation. a** Clonogenic survival of RPE1 parental and PBRM1 knockouts treated with increasing doses of the CDK1 inhibitor RO-3306 relative to DMSO-only treated cells. $n = 6$, mean ± SEM, data were analysed using a 2way ANOVA with Dunnett's test, **$p = 0.0053$, ***$p = 0.0005$. **b** Representative image from clonogenic survival assay in (**a**). **c** Experimental outline for quantifying nuclear defects induced following CDK1 inhibition by RO-3306. **d** % of cells with mild (left) or severe (right, also see panel **f**) nuclear defects after 24 h of treatment with DMSO (-) or RO-3306 (+). $n = 4$, mean ± SEM, data were analysed by 2way ANOVA with Dunnett's test, ****$p < 0.0001$. 600-1500 nuclei were analysed per condition. **e** Representative images from **d**, showing nuclear morphology after 24 h of treatment with DMSO (-) or RO-3306 (+). Scale bar corresponds to 40 µm. **f** Quantification of types of severe nuclear morphology defects in parental or PBRM1 knockout cells treated as per the schematic in **c**. $n = 4$, mean ± SEM. **g** Clonogenic survival of RPE1 parental and PBRM1 knockouts following *CCNB1* siRNA (si1 or si2) treatment, normalised to survival after treatment with a scramble (scr) siRNA. Points correspond to independent biological replicates, $n = 3$, mean ± SEM, data were analysed by 2way ANOVA with Dunnett's test, ***$p = 0.001$ and ***$p = 0.002$ for KO1 and KO2, respectively. **h** Representative image from clonogenic survival assay in **g**. **i** Clonogenic survival of a panel of renal cell carcinoma (RCC) cell lines, which are PBRM1-proficient (blue) or -deficient due to loss-of-function mutations (red). Survival was measured after *CCNB1* depletion with two independent *CCNB1* siRNAs, normalised to survival after treatment with a scramble siRNA. Points correspond to independent biological replicates, $n = 4$ (RCC-FG2, Caki-1, RCC-4), $n = 5$ (786-O), or $n = 8$ (769-P, Caki-2). Boxes contain the 25th to 75th percentiles with line at median, and whiskers extend to 10th and 90th percentiles. **j** Western blotting for PBRM1 in RCC cell lines used in (**i**). α-tubulin is used as a loading control. **k** Representative images in PBRM1-proficient (786-O) and -deficient (RCC-FG2) cell line from the survival assay in (**i**). Source data are provided as a source data file.

Fig. 2e, and Supplementary Fig.14a, b), also do not exhibit sensitivity to CCNB1 depletion (Supplementary Fig. 14c, d). We further tested this using a panel of six renal cancer cell lines, in which three have loss of function mutations in PBRM1, and we found that PBRM1 deficient cells are considerably more sensitive to CCNB1 depletion than the PBRM1 proficient cell lines (Fig. 6i–k, and Supplementary Fig.12d, e), suggesting potential clinical implications. These data suggest that loss of PBRM1 reduces the ability to cope with mitotic perturbations.

### PBRM1 KO cells display sensitivity to Mps1 inhibition in vitro and in vivo

Cells with aberrant centromeres rely on the activity of the spindle assembly checkpoint (SAC) if these changes impact on kinetochore-microtubule attachments[25]. We, therefore, tested whether cells with PBRM1 loss are sensitive to inhibition of the Mps1 kinase that regulates the SAC[25]. Using three different Mps1 inhibitors, we found that RPE1-derived PBRM1 KO cells are modestly but significantly sensitive when compared with the parental cells (Fig. 7a–c, and Supplementary Fig. 13a–c). In contrast, we found no substantial sensitivity to Mps1 inhibitors in the 786-O derived PBRM1 KO cells (Supplementary Fig. 14e–h), supporting the link between downregulation of centromere-associated proteins and perturbation of mitosis.

To test whether this has clinical potential, we created two PBRM1 KO clones in the mouse melanoma B16-F10 cell line for in vivo studies (Fig. 7d and Supplementary Fig. 13d). We first analysed the proteome of these cells by mass spectrometry, and strikingly, we found down-regulation of centromere- and pericentromere-associated proteins (Fig. 7e), suggesting that this is a conserved feature of PBRM1 loss in at least one other species. We next tested the response to Mps1 inhibitors and found the survival of the mouse PBRM1 KO cell lines was significantly lower than the parental B16-F10 cells (Fig. 7f-h, and Supplementary Fig. 13e), consistent with a functional impact of altered centromere-associated proteins.

We used these cell lines to perform in vivo studies with Mps1 inhibitors. Mice were dosed twice weekly with the BOS172722 Mps1 inhibitor over the course of 42 days, at which point the study was terminated (Supplementary Fig. 13f). Mps1 inhibitor treatment had a marginal and statistically insignificant impact on survival of mice bearing B16-F10 parental cell line derived tumours (Fig. 7i and Supplementary Fig. 13g). In contrast, despite the fact the PBRM1-deficient tumours grow more slowly than the parental B16 tumours, survival was further improved following Mps1 inhibitor treatment in mice bearing the PBRM1 KO derived tumours (Fig. 7j, k, and Supplementary Fig. 13g). The disproportionate impact of Mps1 inhibitors on PBRM1-deficient tumours was also evidenced by the fact that tumour growth of PBRM1 KO-derived tumours, but not that of parental B16-F10 derived tumours, was significantly reduced following treatment with Mps1 inhibitors (Fig. 7l-n and Supplementary Fig. 13h). Together, these data

suggest that reliance on the SAC in PBRM1 KO cancers represents a therapeutic vulnerability that can be clinically exploited.

### Loss of PBRM1 leads to centromere fragility

These results raised the possibility that cells lacking PBRM1 display centromere fragility. To test this, we examined the patterns of sister chromatid exchanges (SCEs) in RPE1 parental and PBRM1 KO cell lines in the absence of any perturbations. We found no significant difference in the total number of SCEs in the PBRM1 KO clones compared with the parental RPE1 cells (Fig. 8a). Notably, however, when we scored the location of the exchanges, we found that they were more likely to localise to centromeres in the PBRM1 KO cells (Fig. 8b–d), leading to whole arm exchanges, consistent with an increased vulnerability at centromeres.

To look more directly at centromere fragility, we made use of an assay in which centromeres are labelled in a strand specific manner to identify sister chromatid exchanges and other centromere-specific rearrangements (termed Cen-CO-FISH[12]; Fig. 8e). As a positive control, we depleted CENP-A, which protects centromere integrity[12]. Strikingly, we found that the PBRM1 KO clones showed significantly elevated levels of aberrant centromere signals when compared with the parental RPE1 cells, at a level similar to that seen in CENP-A-depleted cells (Fig. 8f–h). We additionally performed the assay in a second cell line background (1BR3-hTERT) and again find elevated levels of aberrant centromere signals in the PBRM1 KO clones when compared with isogenic controls (Supplementary Fig. 15a, b).

Our mapping data suggests that other SMARCA4-containing complexes are bound at flanking pericentromeric regions (Fig. 3). We were therefore interested to see whether other SWI/SNF complexes are involved in protecting genome stability at centromeres. We performed the Cen-CO-FISH assay in RPE1 cells with CRISPR-mediated KO of the ARID1A subunit of BAF (Supplementary Fig. 15c), which is also commonly mutated in cancer[8]. Notably, in contrast to the effect of PBRM1 loss, we found no detectable difference in aberrant centromere signals in the ARID1A KO cells when compared with parental RPE1 (Supplementary Fig. 15d, e). Collectively, these data indicate that loss of PBRM1 leads to substantially increased genome instability at centromeres, even in the absence of any perturbations.

## Discussion

Here, we find that changes to the structure of centromeres and centromere-associated protein levels are conserved features of PBRM1-deficient cells. We find PBRM1 directs SMARCA4 to specific locations in chromatin flanking centromeres. In the absence of PBRM1, cells show mitotic defects and vulnerability to inhibition of the spindle assembly checkpoint, which has the potential to be therapeutically exploited. Most notably, we find centromere fragility in PBRM1-deficient cells, even in the absence of any perturbation.

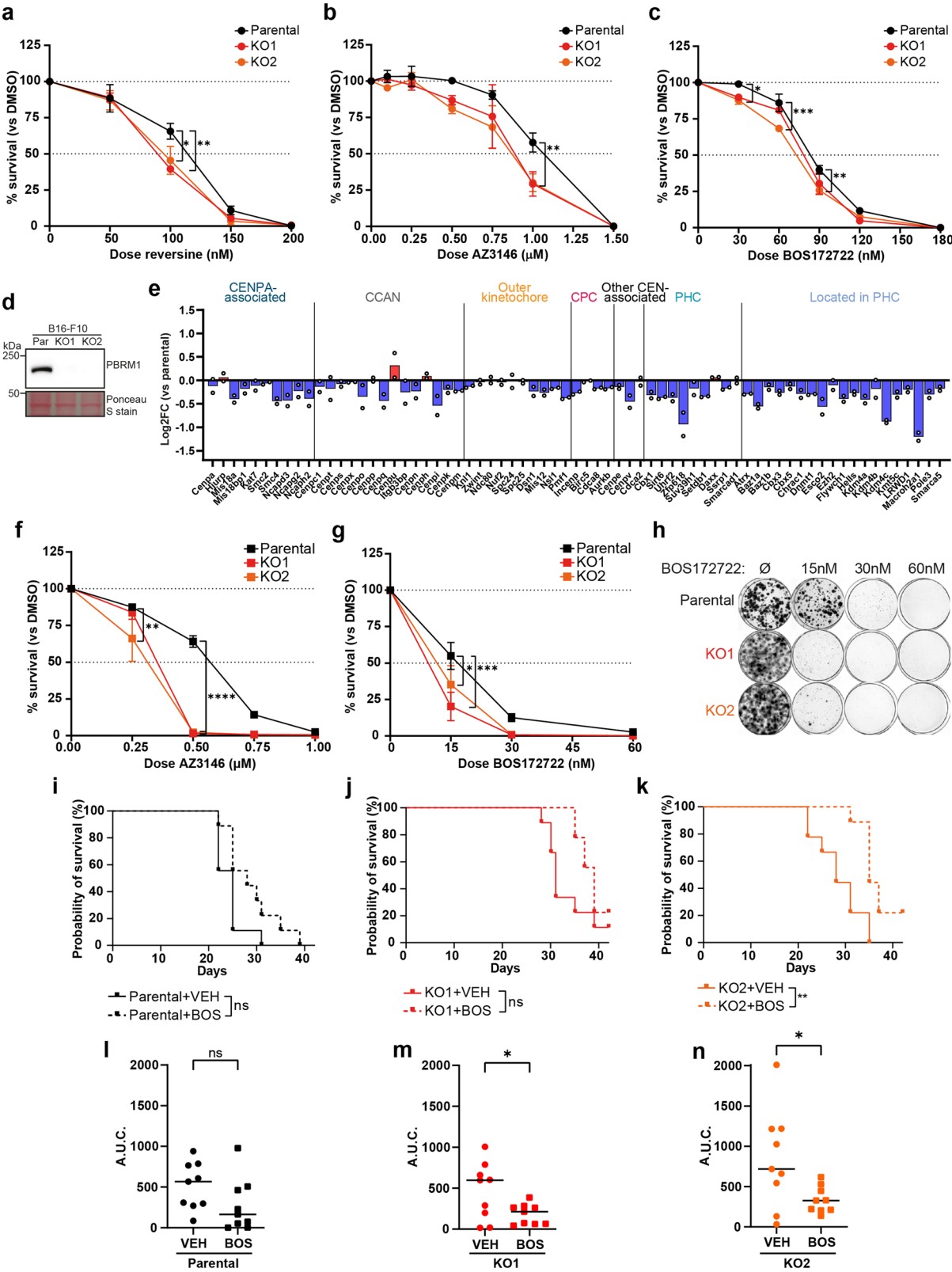

We find several specific changes when PBRM1 is deficient, including altered H3K9 methylation and changes in both levels and staining patterns of CENP-B. Whether there are additional changes to chromatin composition in these regions remains to be tested. Interestingly in that regard, DNA methylation influences CENP-B binding[26] and SWI/SNF can influence DNA methylation[27].

Given the PBRM1-dependent changes in H3K9 methylation patterns, the altered immunofluorescence patterns observed (Fig. 2 and Supplementary Figs 4 and 5) likely arise from altered three-dimensional chromatin structure. However, another possibility is that PBRM1-dependent centromere fragility leads to amplification of repetitive elements, which could result in similar patterns. This

**Fig. 7 | PBRM1 KO cells display sensitivity to Mps1 inhibition in vitro and in vivo. a–c** Clonogenic survival of RPE1 parental and PBRM1 knockout cells after treatment with increasing doses of the Mps1 inhibitors reversine (**a**), AZ3146 (**b**), and BOS172722 (**c**). Data shown are the mean ± SEM of 3 (**b**, **c**) or 4 (**a**) independent biological replicates, and data were analysed by 2way ANOVA with Dunnett's test. In (**a**), *$p = 0.0136$, **$p = 0.0025$; in (**b**), **$p = 0.0042$ and **$p = 0.0056$ for KO1 and KO2, respectively; and in (**c**) *$p = 0.0184$, **$p = 0.0023$, and ***$p = 0.0002$. **d** Western blotting for PBRM1 in B16-F10 parental and PBRM1 knockout cells. Ponceau S staining shows total protein levels. **e** Median Log2FC of annotated pericentromere and centromere proteins using LC-MS of whole cell protein extracts in B16-F10 PBRM1 knockouts compared to parental cells. Points correspond to individual knockout clones from one of two independent biological replicates. **f, g** Clonogenic survival of B16-F10 parental and PBRM1 knockout cells after treatment with increasing doses of Mps1 inhibitors AZ3146 (**f**) and BOS172722 (**g**). $n = 4$

independent biological replicates, mean ± SEM, data were analysed by 2way ANOVA with Dunnett's test. In (**f**), **$p = 0.0043$ and ****$p < 0.0001$; in (**e**), *$p = 0.0405$ and ***$p = 0.0002$. **h** Representative image from clonogenic survival assay in **g**. **i–k** Kaplan-Meier survival curve showing survival of mice with tumours derived from (**i**) B16-F10 parental or (**j**, **k**) PBRM1 knockout cells, treated with vehicle (VEH) or BOS172722 (BOS). $n = 9$ mice per condition, and survival was compared using the logrank (Mantel-Cox) test, **$p < 0.01$. **l–n** Area under the curve (A.U.C.) was calculated based on tumour growth up to the latest timepoint where all mice were still surviving in the indicated cell line; Day 21 for tumours derived from (**l**) B16-F10 parental and (**m**) PBRM1 KO2 cells, and Day 27 for PBRM1 KO1 cells. $n = 9$ mice per condition, and significance was determined using an unpaired two-sided t-test, *$p = 0.0374$ and *$p = 0.0296$ for (**m**) and (**n**) respectively. Source data are provided as a source data file.

possibility seems less likely since it should lead to greater signal heterogeneity than what was detected here, but recombination-mediated instability may exacerbate structural changes over longer timeframes in PBRM1-deficient cells.

Together, these data are consistent with a model (Supplementary Fig. 15f) in which PBRM1-containing PBAF complexes work in the transition arms flanking centromeres to help to create a chromatin substrate that promotes the assembly of protein complexes and structures. In the absence of PBRM1, the integrity of these structures is compromised, leading to mitotic vulnerability, dependence on the spindle assembly checkpoint, and inappropriate recombination between repetitive sequences at the centromere.

When mapping SMARCA4 to pericentromeric regions, we find that a subset of binding sites is lost when PBRM1 is deficient, suggesting that PBRM1 is important for directing SMARCA4-containing complexes to these sites. This raises the possibility that there are acetylated substrates at these sites acting as beacons for the PBRM1 bromodomains.

Perhaps more surprisingly, we find that SMARCA4 localises to new sites when PBRM1 is deficient, and many of these sites are normally enriched in H3K9me3. We also find that H3K9me3 patterns are altered when PBRM1 is deficient and that PBRM1 localisation anti-correlates with H3K9me3 enriched regions. These data are consistent with the idea that PBRM1-containing PBAF complexes help to establish appropriate patterns of H3K9me3, which is important for the spatial organization of chromatin[28]. It was recently reported that DSBs exist in centromeres[6]. We previously showed that PBRM1 influences cohesin dynamics, both at centromeres[10] and at DNA double strand breaks, to help promote repair[29]. H3K9me3 can also influence centromere function and DSB repair[3,30], and there is an interplay between H3K9me3 and cohesin dynamics[31,32]. Therefore, one attractive possibility is that PBRM1 could promote stability at centromeres and pericentromeric regions by modulating the chromatin environment (via H3K9me3 patterns and cohesin dynamics) to prevent inappropriate DSB processing, which would lead to aberrant recombination events between repeats.

Importantly, PBRM1 loss was recently identified as a genetic determinant for a chromosome instability signature that is associated with whole arm or whole chromosome changes[33]. This is consistent with the role in preventing centromere fragility we identify here. Notably, ARID1A, which is frequently mutated in cancer, was not identified as a determinant for chromosomal instability, and consistently, we find no impact of ARID1A on centromere fragility in our assays. This suggests that, while BAF may also be bound in regions in or around centromeres, it does not provide the same function as PBAF complexes to prevent centromere fragility. It is significant that we observe PBRM1-dependent centromere fragility in the absence of any exogenous perturbation, suggesting that this fundamental feature of PBRM1 loss is a critical activity that contributes to tumourigenesis and cancer progression.

## Methods

All research complied with relevant ethical regulations. The in vivo studies were approved by the Animal Welfare and Ethical Review Body' at the Institute of Cancer Research (ICR; protocol number BSU_SPF_1467) and carried out under the UK Home Office Project License P5541E04C.

### Statistics and reproducibility

Sample sizes were decided based on standard practise from the field, and experiments generally had $n = 3$ or $n = 4$ biological replicates (annotated in figure legends). No statistical method was used to predetermine sample size. Biological replicates were used to ensure data was replicable. Three technical replicates were also used where appropriate, i.e. for survival assays, RT-qPCR, proliferation assays. Outliers were removed using Grubbs' test, and only data with a significance level of 0.05 were excluded. A single proteomics biological replicate in 1BR3-hTERT cells was also excluded as it was identified as an outlier based on principal component analysis (PCA). Ranges of doses of drugs and timepoints were also used, as well as multiple assays to confirm results. Multiple knockout clones were also used to ensure observations were biologically reproducible. Proteomic and transcriptomic data were also used to validate results. Randomisation was not used in this paper; rather results were compared to control samples. Blinding was used for manual analyses of clonogenic survival assays and Cen-CO-FISH quantification.

### Cell culture

All cell lines were obtained from ATCC unless otherwise specified. hTERT-RPE1 (catalogue number CRL-4000) cells were cultured in Dulbecco Modified Minimal Essential Medium (DMEM)/F-12 (Sigma) supplemented with 10% FBS (Gibco), 200 μM glutamax (Gibco), 0.26% sodium bicarbonate (Gibco), and 1% penicillin/streptomycin (P/S) (Sigma). 1BR3-hTERT (Gift from Penny Jeggo, Sussex University), HEK293TN (Systems Biosciences LV900A-1), U2OS (catalogue number HTB-96), RCC4-VO (European Collection of Authenticated Cell Cultures, ECACC, catalogue number 03112702), and B16-F10 (catalogue number CRL-6475) cells were cultured in DMEM supplemented with 10% FBS and 1% P/S. 786-O (catalogue number CRL-1932), 769-P (catalogue number CRL-1933), and RCC-FG2 (Cell Lines Service, CLS GmbH, catalogue number 300249) cells were cultured in RPMI 1640 medium (Sigma) supplemented with 10% FBS and 1% P/S. Caki-1 (catalogue number HTB-46) and Caki-2 (catalogue number HTB-47) cells were cultured in McCoy's 5 A modified medium (Gibco) supplemented with 10% FBS and 1% P/S. All cells were maintained at 37°C in a humified incubator with 5% $CO_2$ and were regularly tested for mycoplasma contamination.

### Drug treatments

For clonogenic survival and SRB assays, cells were treated with a range of concentrations of RO-3306 (Merck), reversine (Sigma),

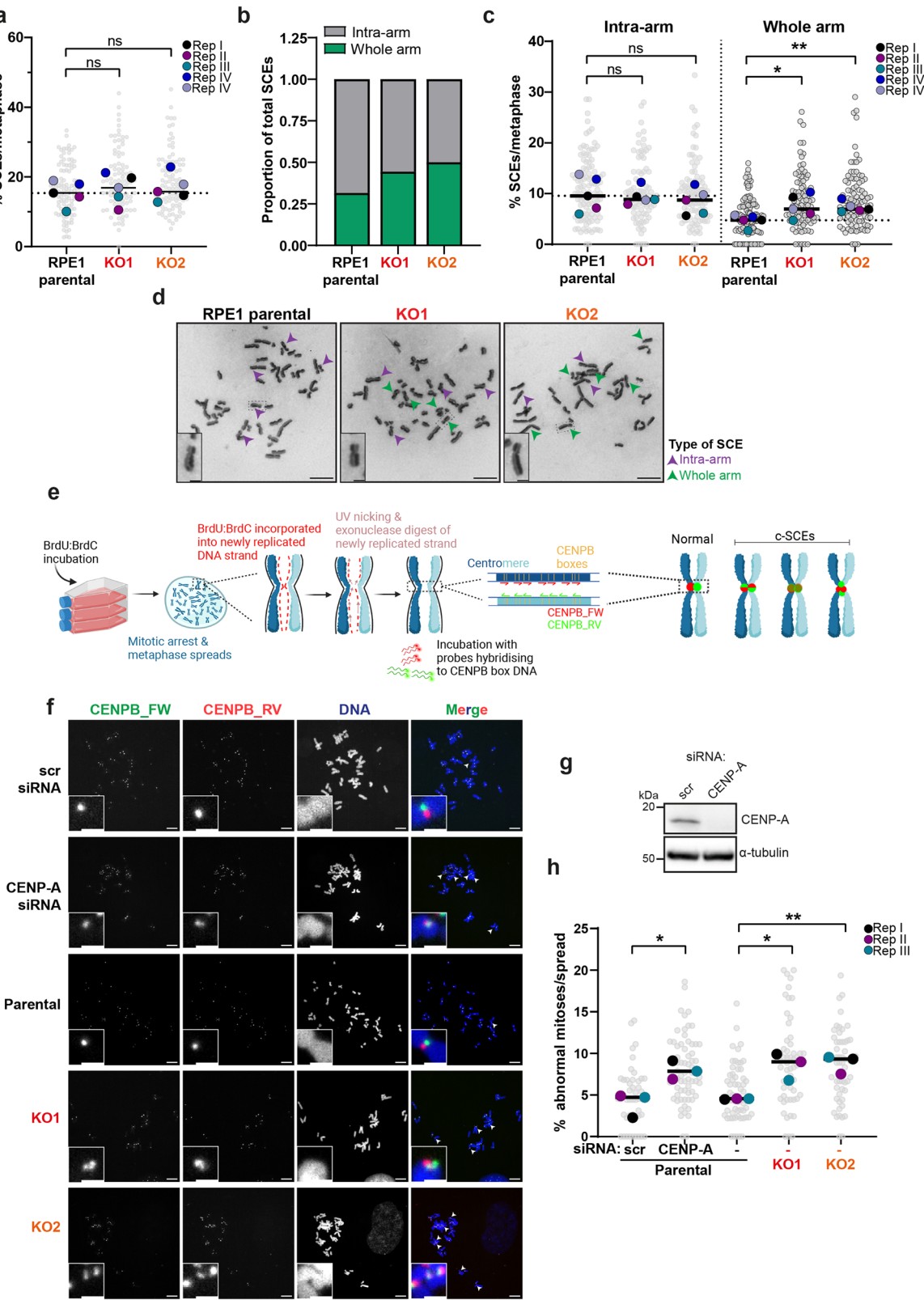

AZ3146 (Sigma), or BOS172722 (MedChemExpress). For immuno-fluorescence experiments, cells were treated with 3 μM RO-3306. At this dose, partial inhibition of CDK1 is seen, perturbing mitotic timing and progression, but not causing the G2 arrest seen at higher doses of RO-3306[34]. For in vivo experiments, BOS172722 ditosylate salt was dissolved in vehicle (10% DMSO and 1% Tween-20 in 89% saline).

## CRISPR/Cas9-mediated gene knockout

For PBRM1 knockout in a range of cell lines and ARID1A knockout in RPE1, sgRNAs (Sigma) targeting PBRM1 or ARID1A Exon 1 were cloned into the pSpCas9(BB)-2A-GFP (PX458) construct. For SMARCA4 knockout in RPE1 cells, two sgRNAs targeting Exon 2 of SMARCA4 were cloned into the CRISPR-Cas9 nickase pSpCas9n(BB)-2A-GFP (PX461) construct. gRNAs are listed in Supplementary Table 1. PX458 and

**Fig. 8 | Loss of PBRM1 leads to centromere fragility. a** Quantification of the % of chromosomes in metaphases with sister chromatid exchanges (SCEs) in RPE1 parental or PBRM1 knockout cells. $n = 5$, grey points indicate individual metaphases, coloured points represent median of each biological replicate, line=median of replicates. At least 90 metaphases per cell line were analysed, two-sided t-test of medians showed no significant difference. **b** Quantification of the proportion of total SCEs which were intra-arm (grey) versus whole-arm (green) exchanges per metaphase, in RPE1 parental and PBRM1 knockout cells. Data are presented as median, $n = 3$. **c** Quantification of the % of chromosomes in metaphases with intra-arm (left) or whole arm (right) exchanges, corresponding to SCE data in **a**, analysed using unpaired two-sided t-test, $*p = 0.0249$, $**p = 0.0035$. **d** Representative images of metaphases in RPE1 parental and PBRM1 knockout cells stained with Giemsa to visualise intra-arm (purple arrows) or whole arm (green arrows) SCEs. Scale bar corresponds to 20 μm. Zoomed images show intra-arm (RPE1 parental) or whole arm (KO1 & KO2) SCEs, scale bar corresponds to 2 μm. **e** Schematic showing Cen-CO-FISH workflow, and methods of quantifying normal versus aberrant

centromeres. Created in BioRender[66]. **f** Representative metaphase spreads from Cen-CO-FISH experiments, with DNA (blue) stained with DAPI, and centromeres hybridised with FISH probes against the CENP-B box (Forward probe, green or Reverse probe, red). White arrows indicate aberrant centromeres. Scale bars correspond to 10 μm. Insets contain zoomed images of representative individual centromeres with scale bars corresponding to 2 μm. **g** Western blotting for CENP-A after treatment of RPE1 parental cells with either scramble or CENP-A siRNA for subsequent Cen-CO-FISH analyses. α-tubulin was used as loading control. **h** Quantification of SCEs at centromeres, defined as the percentage of chromosomes with aberrant centromeres in each metaphase spread (% abnormal mitoses). At least 49 metaphases were quantified for each condition. $n = 3$, grey points represent individual metaphases, coloured points represent median of each biological replicate, line=median of replicates, data were analysed using an unpaired two-sided t-test, $*p = 0.0149$ and $*p = 0.0126$ for CENP-A siRNA and KO1 conditions, respectively, $**p = 0.0027$. Source data are provided as a source data file.

PX461 plasmids were a gift from Professor Feng Zhang (Addgene plasmids #48138 and #48140 respectively). Plasmids were transfected into cells using Lipofectamine 3000 (Invitrogen), according to the manufacturer's instructions. 48 h after transfection, GFP- (PX458) or mRuby-positive (PX461) cells were single-cell sorted into 96-well plates using a FACSAria™ III sorter (BD). Resulting clones were screened for loss of PBRM1 or SMARCA4 using western blotting, followed by Sanger sequencing (Eurofins) of the targeted genomic region, as well as immunofluorescence microscopy and proteomic profiling. The PBRM1 knockouts in RPE1 and 1BR3 were described in ref. 9.

### siRNA mediated gene depletion/RNA interference
100 μM of the specified siRNAs (Dharmacon) were reverse transfected into cells using Lipofectamine RNAiMAX (Invitrogen), according to the manufacturer's instructions. Multiple single siRNAs or pools containing a mix of 4 independent siRNAs were used as indicated in Figure legends. Corresponding single or pooled non-targeting siRNAs (Dharmacon) were used as controls. Cells were assayed 24-72 h after transfection.

### Clonogenic survival assay
Cells in culture were diluted to the appropriate density (300–1000 cells per dish, depending on the cell line) and seeded onto 6 cm dishes in triplicate. For survival after drug treatments, drugs were added 24 h after seeding. For clonogenic survival after RNAi, cells were reverse transfected as described with the appropriate siRNA 24 h before seeding. Cells were incubated in culture for 10-21 days (depending on cell line), after which media was aspirated and colonies were fixed and stained with 1% methylene blue in 70% methanol for 1 h at room temperature with gentle rocking. Dishes were washed extensively with water and allowed to dry overnight. Colony number was counted using a Digital Colony Counter (Stuart), and survival was defined as the % of colonies in treated conditions versus control conditions (vehicle only or control siRNA for drug treatments and RNAi respectively).

### Proliferation assays
For cell growth assays, $1 \times 10^5$ cells were seeded to 6 well plates. Every 24 h, triplicate wells were trypsinised and counted, up to a total of 96 h, to quantify the speed of proliferation. For hTERT-RPE1 cells, cells were seeded in triplicate into 96-well plates, and the rate of proliferation was quantified on the Incucyte SX5 (Sartorius) using phase contrast images of cells taken every 4 h, up to a total of 116 h.

### Detection of dead cells
Cells were seeded in triplicate to clear-bottom black 96 well plates (Greiner Bio-One) and incubated in culture for 48 to 96 h. 0.5 μg/mL Hoechst and 1 μg/mL propidium iodide was added to cells 30 minutes

before detection of total and dead cells using a Celigo S image cytometer (Nexcelom Bioscience).

### Flow cytometry
For flow cytometric analysis of cell cycle distribution, growth medium was collected, and cells were trypsinised and combined with cells suspended in growth medium. Cells were pelleted by centrifugation and washed once with PBS. After removal of most supernatant, cells were gently resuspended in remaining PBS and fixed by addition of 70% ethanol dropwise with gentle vortexing and incubated at -20°C overnight. Cell suspensions were centrifuged for 5 minutes at 300 x $g$ and the ethanol removed. Pellets were washed twice in cold PBS and then resuspended in the appropriate volume of PBS containing 5 μg/mL propidium iodide (Sigma) and 0.1 mg/mL RNase A. Cells were incubated at 37°C for 30 minutes, protected from light. DNA content of at least 10,000 cells per condition was detected on a BD LSR II or BD FACSymphony A5 (BD Biosciences). Cell debris and doublets were removed by gating (Supplementary Fig. 2f) and cell cycle phases were quantified using FlowJo software (v10.8.1).

### Immunofluorescence microscopy
Cells for immunofluorescence (IF) imaging were seeded onto glass coverslips in 6 well plates and assayed. Cells were fixed with 100% ice-cold methanol for 15 minutes at -20°C, followed by rehydration with four 5-minute washes in PBS. Samples were blocked with 1% BSA in PBS for 1 h at room temperature and incubated overnight at 4°C in primary antibody diluted in blocking solution. Following washes in PBS, cells were incubated with the appropriate secondary antibody and DNA stain Hoechst 33342 (Sigma) for 2 h at room temperature. Cells were then washed in PBS, mounted onto frosted glass microscope slides with ProLong Gold (Thermo Fisher Scientific), and cured overnight. Antibodies used for immunofluorescence imaging are indicated in Supplementary Table 2. Cells were imaged with a CSU-W1 Yokogawa Advanced Spinning Disk or CSU-W1 SoRa Yokogawa Super Resolution Spinning Disk Confocal microscope with SlideBook imaging software (3i). 1.02 μm Z-stacks were imaged using a 40x or 63x oil objective, or 0.27 μm Z-stacks when using the SoRa spinning disk, and exported for analysis as maximum intensity projections.

### Fluorescence in situ hybridisation (FISH)
Cells were trypsinised and washed once in PBS. Washed cells were then fixed by adding ice-cold fixative solution (3:1 methanol:acetic acid) dropwise with gentle vortexing. Cells were incubated for 15 minutes at room temperature, centrifuged, and subjected to a second round of fixation. Cells were then resuspended in 30 μL fixative solution, dropped onto a glass microscope slide, and allowed to dry overnight. After drying, cells were rehydrated in 2x sodium chloride and sodium citrate (SSC) buffer for 2 minutes followed by dehydration in an ethanol series

(70, 80, and 95%) for 2 minutes each. The chromosome 2 and chromosome 10 α-satellite FISH probes (Cytocell) were denatured and hybridised with slides according to the manufacturer's protocol. After hybridisation, cells were washed in 0.25x SSC for 2 minutes at 72°C and 2x SSC with 0.05% Tween-20 for 30 seconds at room temperature. Slides were drained and mounted with Vectashield Antifade mounting medium with DAPI (Vector labs).

## Metaphase spreads

Cells in culture were treated with 0.1 μg/mL colcemid (Gibco) for 3-5 h to accumulate cells in mitosis. Cells were trypsinised, saving floating cells and PBS washes in a falcon tube, and the supernatant and resuspended cells were pelleted, washed with PBS, and slowly resuspended in 10 mL of pre-warmed hypotonic solution (75 mM potassium chloride) and incubated for 30 minutes at 37°C with intermittent gentle inversion. 200 μL fresh ice-cold fixative solution (3:1 methanol:acetic acid) was added to solution directly before centrifugation at 175 x $g$ for 5 minutes. The majority of the supernatant was removed, leaving less than 0.3 mL, and the pellet was gently resuspended in the remaining supernatant by tapping. Cells were fixed by adding 10 mL fixative solution dropwise with gentle vortexing and incubated at room temperature for 15 minutes. Cells were washed once with fixative solution and resuspended in the appropriate amount of fixative solution to result in a cell concentration of approximately 1×10$^6$ cells/mL. Drops of approximately 20 μL were dropped onto humid glass slides from a height and air-dried overnight.

## Sister chromatid exchange (SCE) assay

SCE assay was performed as described previously[35,36]. Briefly, cells were labelled for two cell cycles, calculated based on doubling time, with 20 μM BrdU. Metaphases were fixed and dropped onto slides, as described above. After drying, slides were immersed in 10 μg/mL Hoechst-33258 (Sigma) for 20 minutes at room temperature. After rinsing in ddH$_2$O, slides were exposed to 365 nm UV light for 30 minutes, immersed in 2x SSC buffer. Slides were then immersed in preheated 2x SSC at 50°C for 1 hr, before being incubated in 10% Karyo-MAX™ Giemsa stain solution (Thermo Fisher) in Sorenson buffer (pH=6.8) for 30 minutes at room temperature. After incubation, slides were rinsed with ddH$_2$O and allowed to dry, followed by mounting with DPX medium (Sigma). Metaphases were imaged on an RGB colour camera at 100x using 3i imaging software.

## Cen-CO-FISH

Cen-CO-FISH was performed as described[37]. Briefly, cells were labelled for a single cell cycle, calculated based on doubling time, with 7.5 mM 5′-Bromodeoxyuridine (BrdU)(Sigma) and 2.5 mM 5′-Bromodeoxycytidine (BrdC)(MP Biomedicals). Cells were trypsinised and metaphase spreads were prepared as described above, ensuring that metaphase spreads were protected from light when drying. The following day, slides were rehydrated in PBS, and treated with 0.5 mg/mL RNase A (Sigma) for 10 minutes at 37°C. Slides were then stained with 1 μg/mL Hoechst-33258 (Sigma) in 2x SSC buffer. Slides covered in 2x SSC buffer were exposed to 365 nm UV irradiation for 30 minutes using a UV lamp (Analytik Jena). Following UV nicking, DNA was digested using Exonuclease III (Promega) dissolved in the appropriate buffer according to the manufacturer's instructions, twice for 10 minutes each. Slides were washed, dehydrated using an ethanol series (70, 90, and 100%), and dried overnight. The following day, slides were hybridised with peptide nucleic acid (PNA) probes against the CENP-B box (PNABio) diluted to a 50 nM concentration in hybridisation solution (10 mM Tris-HCl (pH=7.4) and 0.5% blocking reagent (Roche) in 70% formamide) and heated to 60°C for 10 minutes directly before use. Slides were first hybridised in a dark hybridisation chamber for 2 h at room temperature with the forward CENP-B probe (PNA Bio, #3004), washed for 30 seconds in Wash Buffer #1 (10 mM Tris-HCl

(pH=7.4) and 0.1% BSA in 70% formamide), and incubated for 2 h with the reverse CENP-B probe (PNA Bio, #F3009). Slides were then washed twice in Wash Buffer #1 for 15 minutes each with gentle rocking, followed by three washes in Wash Buffer #2 (0.1 M Tris-HCl (pH=7.4), 0.15 M NaCl, and 0.1% Tween-20), including 1 μg/mL DAPI (Sigma) in the second wash. Slides were dehydrated in an ethanol series (70, 90, and 100%), air dried, and mounted using ProLong Gold. Slides were imaged as before on an Advanced Spinning Disk confocal microscope but instead imaging 0.2 μm Z-stacks and were exported as maximum intensity projections.

## Whole protein extraction

Cell pellets were resuspended in the appropriate volume of lysis buffer (10% glycerol, 50 mM Tris-HCl (pH=7.4), 0.5% NP-40, 150 mM NaCl) containing 0.25U/μL benzonase nuclease (Sigma), 1x cOmplete™ EDTA-free protease inhibitor cocktail (Roche), 1 mM MgCl$_2$, and 1x PhosSTOP™ phosphatase inhibitor cocktail (Roche). Cells were lysed on ice for 45 minutes followed by centrifugation for 15 minutes at 16,000 x $g$ at 4°C. The resulting supernatant containing whole cell protein extracts was collected. For the SMARCA4 western blot in Supplementary Fig. 7, whole cell extracts were obtained using urea buffer as described previously[38]. Protein concentration was estimated using a Bradford assay (Bio-Rad) according to the manufacturer's protocol.

## Western blotting

For each western blot, approximately 30 μg of protein extract was combined with LSB containing 5% β-mercaptoethanol and boiled for 10 minutes at 95°C to denature proteins. Proteins were separated via SDS-polyacrylamide gel electrophoresis and transferred to 0.2 μm nitrocellulose membranes (Fisher Scientific). Successful protein transfer was confirmed by incubating membranes in Ponceau S solution (Sigma) for 5 minutes with rocking and membranes were blocked in 5% milk in TBS buffer containing 0.1% Tween-20 for 1 h with gentle rocking. Membranes were incubated in the appropriate antibody diluted in blocking buffer at 4°C overnight, washed 3 times with TBS buffer containing 0.1% Tween-20, and incubated in the appropriate horseradish peroxidase (HRP)-conjugated secondary antibody diluted in blocking buffer for 1 h at room temperature (except for β-actin antibody, which is already HRP-conjugated). Proteins were visualised on an iBright CL750 imager (Thermo Fisher Scientific) using Immobilon Forte HRP substrate (Merck Millipore) for chemiluminescence. Details of the primary and secondary antibodies used for western blotting in this study are detailed in Supplementary Table 2.

## RT-qPCR

RNA was extracted from cells using an RNeasy Mini Kit (Qiagen) according to the manufacturer's protocol. 0.5 μg of RNA was then reverse transcribed to cDNA with the High-Capacity cDNA Reverse Transcription kit (Applied Biosystems) according to the manufacturer's protocol. 4 ng of cDNA was used for each qPCR reaction, along with the indicated forward and reverse primers (Supplementary Table 1) at 200 nM concentration, and Power SYBR green PCR master mix (Applied Biosystems). Samples were run in triplicate in MicroAmp Fast Optical 96-well plates (Applied Biosystems) on a StepOne Plus Real-Time PCR system (Applied Biosystems), according to the manufacturer's protocol. cDNA levels were compared using the Ct (comparative cycle) method, and GAPDH and PPIA were used as housekeeping genes to normalise data.

## Proteomics analysis – sample preparation

Cell pellets were lysed in 150 μL buffer containing 1% sodium deoxycholate (SDC), 100 mM triethylammonium bicarbonate (TEAB), 10% isopropanol, 50 mM NaCl and Halt protease and phosphatase inhibitor cocktail (100x) (ThermoFisher Scientific) on ice, assisted with probe

sonication, followed by heating at 90 °C for 5 min and re-sonication for 5 sec. Protein concentration was measured with the Quick Start Bradford protein assay (Bio-Rad) according to the manufacturer's instructions. Protein aliquots of 60 µg or 100 µg (for TMTpro and TMT11plex respectively) were reduced with 5 mM tris-2-carboxyethyl phosphine (TCEP) for 1 h at 60 °C and alkylated with 10 mM iodoacetamide (IAA) for 30 min in the dark, followed by overnight digestion with trypsin at a final concentration of 75 ng/µL (Pierce). Peptides were labelled with the TMT-11plex or TMTpro reagents (ThermoFisher Scientific) according to the manufacturer's instructions. The TMT labelled peptide pool was acidified at 1% formic acid, the precipitated SDC was removed by centrifugation, and the supernatant was SpeedVac dried. Peptides were fractionated with high-pH Reversed-Phase (RP) chromatography with the XBridge C18 column (2.1 x 150 mm, 3.5 µm) (Waters) on a Dionex UltiMate 3000 HPLC system. Mobile phase A was 0.1% (v/v) ammonium hydroxide and mobile phase B was acetonitrile, 0.1% (v/v) ammonium hydroxide. Peptides were fractionated at a flow rate of 0.2 mL/min using the following gradient: 5 min at 5% B, for 35 min gradient to 35% B, gradient to 80% B in 5 min, isocratic for 5 minutes and re-equilibration to 5% B. Fractions were collected every 42 sec, combined in 28 fractions and SpeedVac dried.

## Proteomics analysis – LC-MS and protein quantification

LC-MS analysis was performed on a Dionex UltiMate 3000 UHPLC system coupled to the Orbitrap Lumos Mass Spectrometer (ThermoFisher Scientific). Peptides were loaded onto the Acclaim PepMap 100, 100µm × 2 cm C18, 5µm, trapping column at 10 µL/min flow rate. Peptides were analysed with the EASY-Spray C18 capillary column (75µm × 50 cm, 2µm). Mobile phase A was 0.1% formic acid and mobile phase B was 80% acetonitrile, 0.1% formic acid. Peptides were separated over a 90 min gradient 5%-38% B at a flow rate of 300 nL/min. Survey scans were acquired in the range of 375-1,500 m/z with a mass resolution of 120 K. Precursors were selected in the top speed mode in cycles of 3 sec and isolated for CID fragmentation with quadrupole isolation width 0.7 Th. Collision energy was 35% with AGC 1e4 and max IT 50 ms. Quantification was obtained at the MS3 level with HCD fragmentation of the top 5 most abundant CID fragments isolated with Synchronous Precursor Selection (SPS). Quadrupole isolation width was 0.7 Th and collision energy was 65%. The HCD MS3 spectra were acquired for the mass range 100-500 with 50 K resolution. Targeted precursors were dynamically excluded from further fragmentation for 45 seconds with 7 ppm mass tolerance. The mass spectra were analysed in Proteome Discoverer 2.2 or 2.4 (ThermoFisher Scientific) with the SequestHT search engine for protein identification and quantification. The precursor and fragment ion mass tolerances were set at 20 ppm and 0.5 Da respectively. Spectra were searched for fully tryptic peptides with maximum 2 missed cleavages and minimum length of 6 amino acids. TMT6plex or TMTpro at N-terminus/K and Carbamidomethyl at C were selected as static modifications. Oxidation of M and Deamidation of N/Q were selected as dynamic modifications. Peptide confidence was estimated with the Percolator node and proteins were identified with at least one peptide with q-value < 0.01 based on target-decoy database search. All spectra were searched against reviewed UniProt Homo sapiens protein entries. The reporter ion quantifier node included a TMT11plex or TMTpro quantification method with an integration window tolerance of 15 ppm at the MS3 level. Only unique peptides were used for quantification, considering protein groups for peptide uniqueness. Only peptides with average reporter signal-to-noise>3 were used for protein quantification.

## Statistical analysis of proteomics data

Log2 ratios against the parental (PBRM1 WT) cells were computed for each PBRM1 knockout (KO) clone, per cell line, using the sum-normalised abundances exported from Proteome Discoverer. For visualising effects on each cell line (Fig. 1g-i and Supplementary

Fig. 2d,e), the median log2(KO/WT) was calculated from replicate clones or biological repeats and was further normalised by subtracting column median; p-values were obtained with two-sided unpaired one-sample t-test in the Perseus platform v1.6.2.2[39]. The following number of replicates were used: RPE1 × 3 clones vs 2 × Parental, × 2 replicates each; 1BR3 × 6 clones vs 1 × Parental; HEK293TN × 3 clones vs 1 × Parental; U2OS × 2 clones vs 2 × Parental, × 2 replicates each; 786 O × 4 clones vs 2 × Parental, × 2 replicates each. Proteins detected in least one parental and PBRM1 knockout cell line were used for GSEA analysis (v4.3.2)[40]. The log2 ratio of classes was used as the metric for ranking genes, and normalised abundances from proteomic data from all 5 cell lines was used as input. Hallmarks and Gene Ontology: Biological Processes gene set collections were used to detect enriched gene sets. Hallmark gene sets enriched in at least 3 out of 5 cell lines (in the same direction, negatively or positively) and with an FDR < 0.25 were included, and the top 20 negatively enriched Biological Processes were plotted. Cancer cell line proteomic data was downloaded from the DepMap portal and analyses were performed on the normalised protein quantitation data of 375 cell lines performed as described previously[41]. Proteins which were not detected in at least 50% of the profiled cell lines were removed from the analysis, and cell lines were ordered based on PBRM1 Log2FC. Pearson correlation was used to determine the correlation between the abundance of PBRM1 and the listed proteins. Heatmaps were generated in R using ComplexHeatmap (v2.14.0)[42] and clustering was performed using default Euclidean distance methods.

## In vivo studies

$2 \times 10^5$ B16-F10 cells (parental or PBRM1 KOs) were resuspended in 100 µL PBS and injected subcutaneously into the right flank of 6-8 week old female C57BL/6 J mice (Charles River Laboratories). After 3 days growth, mice were treated with vehicle or 50 mg/kg BOS172722 via oral gavage. Mice were treated twice weekly (Monday and Thursday) until the experiment endpoint (Day 42) or were culled once the tumours reached 15 mm size in any direction (maximum permitted size, which was never exceeded). Housing conditions were 21°C +/- 2°C at 45-65% humidity with 12 h/12 h light/dark cycles.

## CUT&RUN sequencing

CUT&RUN (Cleavage Under Targets & Release Using Nuclease) was performed according to the CUT&RUN Assay Kit protocol (Cell Signalling Technology) with the following modifications. Cells were detached with Accutase (Sigma). $2 \times 10^5$ cells were collected per experiment and pelleted by centrifugation for 5 minutes at 600 x g. Beads were incubated in the indicated primary antibody (Supplementary Table 2) on a nutator overnight at 4 °C. MNase digestion was carried out at 0 °C in an ice water bath for 30 minutes. Salt fractionation and DNA purification were carried out according to the CUT&RUN.salt protocol[17]. Briefly, after STOP buffer addition, samples were incubated at 4 °C for 1 h, and the supernatant containing the low salt fraction was collected. Beads were resuspended in low salt buffer (175 mM NaCl, 10 mM EDTA, 2 mM EGTA, 0.1% TritonX-100, 20 µg/mL glycogen). High salt buffer (825 mM NaCl, 10 mM EDTA, 2 mM EGTA, 0.1% TritonX-100, 20 µg/mL glycogen) was added to beads dropwise with gentle vortexing. Samples were again rocked at 4 °C for 1 h, centrifuged for 5 minutes at 16,000 x g, and the supernatant containing the high salt fraction was collected. Low salt fractions were adjusted to 500 mM NaCl. RNAse A (Thermo Fisher Scientific) was added to all fractions and samples were incubated at 37 °C for 20 minutes. SDS (sodium dodecyl sulphate) and Proteinase K (Cell Signaling Technology) were added and samples were incubated at 50 °C for 1 h. DNA was extracted using phenol/chloroform and precipitated in 100% ethanol as described[17] before library preparation. Libraries were prepared using the NEBNext Ultra II DNA library prep kit for Illumina (New England Biolabs), profiled using the Agilent TapeStation D1000 high

sensitivity ScreenTape on the Agilent 4150 TapeStation System, and sequenced on the Illumina Novaseq 6000 with 150×150 bp reads by Novogene (Novogene Corporation Inc. UK). For PBRM1 knockout cells, SMARCA4 replicates 1 and 2 were performed with KO1 and SMARCA4 replicate 3 was performed with KO2. H3K9me2/3 replicates 1 and 3 were performed with KO1 and replicate 2 with KO2.

## CUT&RUN analysis

Fastq reads were trimmed using TrimGalore (v0.6.6) (Krueger F, Trimgalore (2023), GitHub repository, https://github.com/FelixKrueger/TrimGalore) using the options --trim-n --paired. The human CHM13-T2Tv1.1 (GenBank GCA_009914755.3) and S. cerevisiae S288C (GenBank GCA_000146045.2) assemblies were combined into one FASTA file, then trimmed reads were mapped using Bowtie2 (v2.4.2)[43] using the parameters --local --very-sensitive-local --no-unal --no-mixed --no-discordant --dovetail --soft-clipped-unmapped-tlen --non-deterministic --phred33 -I 50 -X 1500. Reads with more than 3 mismatches were removed with Sambamba (v0.5.0)[44], and their corresponding mates were removed with Picard tools (v2.23.8) ("Picard Toolkit." 2019. Broad Institute, GitHub Repository. https://broadinstitute.github.io/picard/; Broad Institute). Sam files were converted to bam with SAMtools (v1.11)[45]. Reads mapped to the CHM13-T2Tv1.1 assembly were extracted using BAMtools (v2.5.1)[46] split. Duplicate reads were removed using Picard if deemed necessary. Low and high salt BAM files from each sample were merged, sorted, and indexed using SAMtools.

## Uniquely mapped reads analysis

Uniquely mapping reads were extracted with Sambamba[44], using the parameters: view -h -t 6 -f bam -F "[XS] == null and not unmapped and not duplicate" Bigwig files were made using Deeptools (bamCoverage --extendReads --maxFragmentLength 1500 --binSize 10 --outFileFormat bigwig, v3.1.3)[47] and normalised by total number of mapped reads, including non-uniquely mapping ones (using --scaleFactor parameter). Peak calling was performed on uniquely mapping read bam files using MACS2 (v2.2.7.1)[48] callpeak using IgG as control, with options -g 3054832041 -f BAMPE --keep-dup all -q 0.01 --broad --broad-cutoff 0.01. Bedtools (v 2.29.2)[49] multiinter and merge functions were used to merge peaks that were called in at least 2 reps. Screenshots of bigwig files and peaks were taken from IGV (v2.11.9)[50]. The ChIPseeker R package (v1.40.0)[51] was used to analyse the genomic distribution of peaks across features. The intervene package (v0.6.5)[52] was used to calculate overlapping peak intersections and resulting venn diagrams were re-drawn using the ggVennDiagram R package (v1.5.2)[53]. The EnrichedHeatmap R package (v1.34.0)[54] was used to generate heatmaps of CUT&RUN and ChIP-seq signal across regions extended +/-5kb around target peaks, with an averaged signal plot above.

## Analysis of downloaded publicly available ChIP-seq datasets

Histone mark ChIP-seq data for RPE1 cells across the cell cycle, and associated input data was downloaded from GEO accession GSE175752[55]. Fastq files across the cell cycle (G1, ES, LS, G2) in RPE.Ctrl cells were merged for downstream analysis. ChIP-seq data of PBRM1 (SRR12036678, SRR23588908), SMARCA4 (SRR12036684, SRR23588912) and input (SRR12036698) in parental 786-O cells, as well as PBRM1 (SRR12036715, SRR23588903) and input (SRR12036716) from A498, and PBRM1 (SRR12036717-SRR12036722) and input (SRR12036723- SRR12036728) from HK2 were downloaded from the GEO accession GSE152681[21]. Fastq files, where multiple SRA files were available for the same sample, were merged for downstream analysis. ChIP-seq data of SMARCA4 (SRR2133615) and input (SRR2133624) from parental HCT116 cells were downloaded from the GEO accession GSE71510[22]. These ChIP-seq datasets were analysed as above for the CUT&RUN with a few exceptions, as they have single-end reads rather than paired-end reads, the Picard tools step to remove corresponding mates following removing reads with 3

mismatches was no longer required. Bowtie2 parameters were changed to -I 300 -X 700 for analysing GSE152681. Following uniquely mapping read filtering, bigwig files were generated with updated bamCoverage parameters to not include --maxFragmentLength, but to update to --extendReads 250 (for GSE175752), --extendReads 500 (for GSE152681) and --extendReads 180 (for GSE71510).

## Read filtering steps for *k*-mer analysis

The seqkit (v2.5.1)[56] fx2tab function was used to identify read IDs of CUT&RUN reads that mapped to the spike-in, S. cerevisiae S288C (converted bam file to fastq using bedtools bamtofastq (v 2.29.2)[49]. The BBmap (v38.84) (BBMap – Bushnell B. – sourceforge.net/projects/bbmap/) filterbyname.sh tool was used to filter out and exclude reads mapping to the S. cerevisiae S288C genome from the original fastq files. CUT&RUN reads mapped to the CHM13-T2Tv1.1 assembly, following extraction of S. cerevisiae S288C reads, were used to calculate the total base count for each replicate for normalisation purposes, following deduplication, trimming and interleaving of paired reads (described below). Reads which intersected with the centromere and pericentromere were extracted for full *k*-mer analysis. Briefly, sorted bam files were converted to paired end bed files using bedtools (v 2.29.2)[49] bamtobed -bedpe function, and then reads that mapped to the centromere and pericentromere were extracted using bedtools intersect function with the CHM13-T2Tv1.1 cenSatAnnotation.bed file. Intersecting read names were printed and used with the BBmap (BBMap – Bushnell B. – sourceforge.net/projects/bbmap/) filterbyname.sh tool to filter to only include centromere and pericentromere mapping reads from the original fastq files. These reads were then used for the full *k*-mer analysis.

*k*-mer analysis pipeline. The (reference-free) *k*-mer analysis pipeline was adapted from previously described methods[1,23]. First, the clumpify.sh tool from BBmap was used to remove duplicate paired reads that could be PCR duplicates. Then, trimmomatic (v0.39)[57] was used to trim adapters from paired reads, using the following parameters PE -phred33 ILLUMINACLIP: TruSeq3-PE.fa:2:30:12 SLIDINGWINDOW:10:10 MINLEN:71. The reformat.sh tool from BBmap was used to interleave paired reads for subsequent analysis. Then the kmc package (v3.2.1)[58] and kmc tools command were used to count and dump 51 bp *k*-mers that occur at least twice in each replicate, using the parameters -k51 -sm -ci2 -cs100000000. The R package dplyr (v1.1.0)(Wickham H, François R, Henry L, Müller K, Vaughan D (2023). _dplyr: A Grammar of Data Manipulation_) was used to 'full_join' the *k*-mer counts of all replicates and any missing values were replaced with a value of 1 (as missing values could have a count of 0 or 1, so to underestimate enrichment, and to allow fold change calculation). *K*-mer counts were divided by the total base counts for each replicate (from the total genome reads) to calculate the normalised *k*-mer counts. Low and high salt fastq files were analysed separately and their normalised *k*-mer counts were added together for downstream analysis. *K*-mers were filtered to only include those with a normalised *k*-mer count of >5e-09 for each replicate individually, and then fold changes were calculated, comparing one sample replicate against the averaged IgG normalised *k*-mer count as a control (e.g. SMARCA4 vs average IgG). Note that 9 IgG replicates were included in the average from each experiment (n = 6 from parental RPE1 cells, and n = 3 from PBRM1 knockout cells). Tidyverse (v2.0.0) (doi:10.21105/joss.01686) and tibble (v3.2.0) (Müller K, Wickham H (2022). _tibble: Simple Data Frames_) packages were also used above for data formatting purposes, using r-base v4.2.3.

## Enriched *k*-mer selection and downstream analysis

Enriched *k*-mers were selected if they had a fold change of >2 in the normalised *k*-mer count of at least 2 replicates vs the averaged IgG normalised *k*-mer count. Enriched *k*-mer output files were converted to FASTA format, using sed '1 d' and awk -F '[]' 'BEGIN{{OFS = "\n"}}{{n=NR;

x = ">"n; print x, $1}}'. Bowtie2 (v2.4.2)[43] was used to map these *k*-mers back to the CHM13-T2Tv1.1 genome, using the specific parameters -f -k 5000 --score-min C,0,0 to allow only exact sequences to be mapped back up to 5000 times. Bedtools (v 2.29.2)[49] intersect was again used to extract *k*-mer mapping sites that intersect with the centromere annotation, and awk -F ["_","\t"] 'OFS = "\t" '¹' was used to split the centromere annotation in the subsequent bed file from e.g. ct_1_1(p_arm) to ct. The GNU datamash package (v1.1.0) (Free Software Foundation, I. (2014). GNU Datamash. Retrieved from https://www.gnu.org/software/datamash/) was then used to create a pivot table summarising how many times each *k*-mer mapped to each part of the centromere annotation (following the split described above), using the following script: sort -k 1 | datamash -s crosstab 1,2 | sed 's ~ N/A ~ 0-g'. Heatmaps representing centromere distribution of *k*-mers were generated in R using ComplexHeatmap (v2.20.00). Bigwig files were generated from enriched k-mer bam files using deepTools bamCoverage tool using the following parameters --smoothLength 1 --binSize 1 --scaleFactor 1 and screenshots were taken from IGV[50]. Motifs associated with enriched *k*-mers were discovered using the MEME-suite SEA *k*-mer enrichment package[59]. Venn diagrams were created with the ggVennDiagram R package[53]. Bedtools (v 2.29.2) reldist function[49] (based on[24]) was used to calculate the relative distance between a query set (*e.g.* H3K9me3 *k*-mer mapping locations) and a reference set (e.g. PBRM1-specific *k*-mer mapping locations). This was calculated using the following formula: min(d1,d2)/r – where d1 and d2 are the distances between each location of the query set and the two closest locations of the reference set, and r is the total distance between the two closest locations of the reference set, i.e. (d1 + d2). The cumulative fractions of query *k*-mer mapping locations at each relative distance up to a maximum possible of 0.5 were calculated and plotted.

## RNA-sequencing

For RNA-sequencing, pellets were harvested by scraping cells in ice-cold PBS following by centrifugation at 7500 x g for 10 minutes. Three independent biological replicates were used for parental cells (n = 3), and two independent biological replicates of two individual PBRM1 knockout clones were used for PBRM1 knockouts, which were combined for n = 4 in total. Pellets were resuspended in 500 μL TRIzol reagent and total RNA was extracted using a Direct-zol RNA miniprep kit (Zymo Research) according to the manufacturer's protocol. RNA concentration and quality was confirmed with the Agilent High Sensitivity RNA ScreenTape, using the Agilent Tapestation as above. Library preparation and sequencing was performed by Novogene Corporation Ltd. Novogene NGS Stranded RNA Library Prep Set was used to generate 250-300 bp insert strand specific libraries, and ribosomal RNA was removed using TruSeq Stranded Total RNA Library Prep. 50 million 150 bp paired-end reads were sequenced on an Illumina NovaSeq 6000. Fastq reads were checked using FastQC (v0.11.9) (Andrews, S. (2010). FastQC: A Quality Control Tool for High Throughput Sequence Data [Online]. http://www.bioinformatics.babraham.ac.uk/projects/fastqc/) and trimmed using TrimGalore (v0.6.6) (Krueger F, Trimgalore (2023), GitHub repository, https://github.com/FelixKrueger/TrimGalore). Residual ribosomal RNA reads were removed using Ribodetector with -e norRNA setting (v0.2.7)[60] and strandedness was detected using RSeQC (v4.0.0)[61]. Reads were aligned to the T2T-CHM13v2 genome using STAR alignment software (v2.7.6a)[62] and reads mapping to genes were quantified using HTSeq-Count (v0.12.4)[63]. Differential analysis of gene expression was calculated in R using DESeq2 (v1.38.3)[64].

## Software & statistical analyses

The number of biological repeats for each experiment and the statistical analyses used are indicated in the figure legend for each experiment. Graphs were generated and statistical analyses were performed using GraphPad Prism (v9.5.1). Outliers were removed using Grubbs' test using a significance level of 0.05. Microscopy images were analysed using ImageJ (v1.5.3 or 1.5.4) or CellProfiler (v4.0.7) software, and visualized using GraphPad Prism or ggplot2 (v3.4.4)(Valero-Mora (2010). ggplot2: Elegant Graphics for Data Analysis. https://doi.org/10.18637/jss.v035.b01). For statistical analyses of microscopy data, the median of each biological repeat was used for the appropriate statistical test, described in the corresponding Figure legend). Omics data were visualised using the indicated packages in R (v4.2.1, 4.2.2, or 4.4.0 except where noted) and R Studio (v2021.09.0, 2023.03.1, 2024.04.2 except where noted). Figures were generated using Adobe Illustrator (v27.5). Graphics in Figs. 1a, j, 2f, and Supplementary Figure 15f were generated using Adobe Illustrator. The schematic in Fig. 8e was generated using BioRender (biorender.com). More details of script and packages used are available at https://github.com/Downs-Lab/PBRM1_centromere.

## Reporting summary

Further information on research design is available in the Nature Portfolio Reporting Summary linked to this article.

## Data availability

The RNA-seq datasets generated in this study are available in the GEO repository (accession number GSE235342; https://www.ncbi.nlm.nih.gov/geo/query/acc.cgi?acc=GSE235342). The CUT&RUN datasets generated in this study are available in the GEO repository (accession number GSE235294; https://www.ncbi.nlm.nih.gov/geo/query/acc.cgi?acc=GSE235294). The proteomics datasets generated in this study are available through the ProteomeXchange Consortium via the PRIDE repository (accession number PXD043209; https://www.ebi.ac.uk/pride/archive/projects/PXD043209). All cell lines used in this study have been made available at CancerTools.org. All data generated in this study are provided in the Supplementary Information figures and tables. Source data are provided with this paper.

## Code availability

The scripts and packages used are described in the methods and available at https://github.com/Downs-Lab/PBRM1_centromere and https://zenodo.org/records/14604006[65].

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

## Acknowledgements

We thank the Institute of Cancer Research Flow Cytometry, Light Microscopy and the Biological Services Unit for support, and we thank the ICR Scientific Computing Team for HPC services. We thank all members of the Downs lab, Jon Pines, and Bill Earnshaw for helpful discussions, Swen Hoelder and Florence Raynaud for reagents and advice, and former lab members Peter Brownlee and Cornelia Meisenberg for early observations. This work was supported by Cancer Research UK C7905/A25715 (JAD), C309/A25144 (AAM), and DRCRPGTD-Nov21100001 (JCS) and the Medical Research Council MR/W001276/1 (JAD, AAM, and JCS).

## Author contributions

K.A.L., A.H., and J.A.D. conceived the project. K.A.L., A.H., L.W., T.I.R., H.F., S.F., K.B., N.A. and F.S. designed and performed experiments. A.H., L.W. and T.I.R. performed bioinformatics analyses. F.T.Z., A.A.M., J.C.S., and J.A.D. obtained funding and provided input and supervision. K.A.L., A.H., and J.A.D. prepared the original draft, and all authors contributed to review and editing.

## Competing interests

The authors declare no competing interests.
