## [Peer Review file · Nature Communications]

PBRM1 directs PBAF to pericentromeres and protects centromere integrity

Corresponding Author: Professor Jessica Downs

Version 0:

Reviewer comments:

Reviewer #1

(Remarks to the Author)

In this manuscript titled "PBRM1 directs PBAF to pericentromeres and protects centromere integrity", the authors presented new findings about the role of the PBAF subunit PBRM1 in the maintenance of centromere integrity and proper chromosome segregation during cell division. They utilized immortalized normal and tumor cell lines with CRISPR-based PBRM1 knock-out models to investigate cell proliferation, morphology, and centromeric organization. They used a multi-omic approach to identify genomic binding and the transcriptomic and proteomic changes in the absence of PBRM1, along with the effect of PBRM1 deficiency in cell proliferation and in vivo tumor growth with mitotic perturbation. They concluded that PBRM1 is essential in maintaining centromeric protein assembly and overall structure, preventing centromeric fragility. Without PBRM1, cells exhibit increased susceptibility to cell cycle checkpoint inhibitors, which may be clinically significant. Overall, the changes in cell dynamics and genomic interaction properties of PBRM1 were presented in detail. However, some concerns need to be addressed.

1. It is interesting that the changes in transcript levels are not reflected in the protein levels of the centromere-associated proteins. Is it similar in all other transcript and protein combinations? A more detailed explanation regarding this mismatch is warranted.
2. Do the centromeric and pericentromeric protein structures form normally without PBRM1? Immunolocalization of representative proteins from each group would help show the status of centromere-associated structures.
3. While CENP-A level and distribution did not change, the alpha satellite DNA area increase needs explanation. Fig. 2a-2b and Fig. S4a-S4b need statistical tests to make a comparative statement. Additionally, the merged panels in Fig. 2C do not match the panels on the left.
4. It was inferred that PBRM1 regulates patterns of H3K9me3-enriched heterochromatic regions, as PBRM1-KO specific peaks of SMARCA4 are normally enriched at heterochromatic regions enriched in H3K9me3. A minor increase in overall H3K9me3 was also observed in the mutants, raising the question of whether the distribution of the increased H3K9me3 in PBRM1 KO cells is shifted and whether that explains the altered occupancy of SMARCA4. The chromatin occupancy of H3K9me3 in PBRM1 mutant cells should be determined to establish the relationship between PBRM1 and H3K9me3.
5. Based on the GSEA results, comparing the frequency of apoptotic cells in PBRM1 mutant cell lines to their parental lines will also be helpful.
6. While the authors have concluded that PBRM1-KO cancers rely on the activity of spindle assembly checkpoint for survival (lines 299-301), based on the tumor volume in Fig. 6J and S11J (statistics absent), both parental and PBRM1 KO B16-F10 cells seem to have responded similarly to SAC inhibition. A major tumor volume reduction is also apparent in vehicle-treated PBRM1 KO cells compared to vehicle-treated parental cells. Therefore, the observed reduction in tumor growth and subsequent survival seems to be due to the loss of PBRM1. Additionally, the same data were plotted in both Fig. 6i and Fig. S11g-l, which should be noted in the corresponding legends.
7. In lines 301-303, authors implied a clinical significance of SAC dependence of PBRM1 KO cancers, which is confusing, as it is clear PBRM1 KO is itself a vulnerability in tumor cells. If they meant PBRM1 mutated tumor, testing such cell lines with the MPS1 inhibitor might be necessary to suggest this clinical significance.
8. In lines 111-112, the statement on the correlation between the PBRM1 level and the level of other centromeric proteins needs to be supported by a correlation metric. Additionally, the groups of centromeric proteins could be clustered together to visualize the relationship better.

Reviewer #2

(Remarks to the Author)

Centromeres are essential for chromosome transmission, but are highly susceptible to DNA breaks. The mechanisms that protect centromere integrity are not fully understood. Lane et al. report that (peri)centromere abnormalities are the common feature of cells deficient in PBRM1, a component of the PBAF chromatin remodelling complex, and propose that this is mediated by its role in organising (peri)centromere chromatin structure. The work is timely and has significant implications for understanding cancer associated with PBAF deficiency.

The work has been meticulously conducted in a variety of conditions where PBRM1 is defective using multiple approaches including proteomics, mapping and scoring of short reads on (peri)centromeres, microscopy, and survival of cell lines and mouse models with appropriate negative controls. It seems to me that the results consistently support their model and the sensitivity of PBRM1-deficient cells/animals to mitotic inhibitors is convincing. I congratulate the authors for obtaining the results from this rather complex study for a very plausible model. I enjoyed reading this manuscript.

The only minor point that was unclear to me was the relationship between the reduced protein levels of centromere-associated factors and PBRM1 deficiency (Figure 1). This is unlikely to be related to altered gene expression (Figure 1I), indicating the alternative possibility of protein stability. Would the centromere assembly of these factors affect their protein stability? For example, CENPB protein levels appear to decrease (Figure 1K, Figure S3a), whereas CENPB binding to centromeres has been shown to be blocked by the CENPB-box methylation (PMID: 15634350). I wonder if PBRM1 is related to such DNA modification. A discussion on this aspect could be very helpful to complete this work.

Reviewer #3

(Remarks to the Author)

Version 1:

Reviewer comments:

Reviewer #1

(Remarks to the Author)

This is an interesting story, and the authors have done a good job of addressing our concerns in the body of the manuscript.

(Remarks on code availability)

Reviewer #2

(Remarks to the Author)

I'm fully satisfied with the revision and recommend the paper for publication in Nature Communication.

(Remarks on code availability)

Reviewer #3

(Remarks to the Author)

(Remarks on code availability)

Thanks to all of the reviewers for their feedback. We have revised the manuscript to take these suggestions into account, and the manuscript is much improved as a result. Specific changes are outlined below.

Reviewer #1 (Remarks to the Author):

In this manuscript titled “PBRM1 directs PBAF to pericentromeres and protects centromere integrity”, the authors presented new findings about the role of the PBAF subunit PBRM1 in the maintenance of centromere integrity and proper chromosome segregation during cell division. They utilized immortalized normal and tumor cell lines with CRISPR-based PBRM1 knock-out models to investigate cell proliferation, morphology, and centromeric organization. They used a multi-omic approach to identify genomic binding and the transcriptomic and proteomic changes in the absence of PBRM1, along with the effect of PBRM1 deficiency in cell proliferation and in vivo tumor growth with mitotic perturbation. They concluded that PBRM1 is essential in maintaining centromeric protein assembly and overall structure, preventing centromeric fragility. Without PBRM1, cells exhibit increased susceptibility to cell cycle checkpoint inhibitors, which may be clinically significant. Overall, the changes in cell dynamics and genomic interaction properties of PBRM1 were presented in detail. However, some concerns need to be addressed.

We thank this reviewer for the detailed feedback and useful suggestions. Changes to the manuscript are outlined in detail below.

1. It is interesting that the changes in transcript levels are not reflected in the protein levels of the centromere-associated proteins. Is it similar in all other transcript and protein combinations? A more detailed explanation regarding this mismatch is warranted.

Thank you for highlighting this. In fact, the lack of correlation between altered transcript and protein levels is a general feature of the PBRM1 deficient cells, and we find very few pathways that show a clearly correlated relationship. This is consistent with previous studies in SWI/SNF-deficient colorectal cancer cells (PMID:28854368, see Figure 4), and it suggests that SWI/SNF activity often influences protein stability through mechanisms other than direct regulation of transcription. As suggested, we have expanded the text in the revised manuscript to highlight this point.

2. Do the centromeric and pericentromeric protein structures form normally without PBRM1? Immunolocalization of representative proteins from each group would help show the status of centromere-associated structures.

This is a good question. To address this, we examined the patterns of CENPB and NDC80 by immunofluorescence and measured the area and shape of the resulting signals in parental and PBRM1 KO cells. We also mapped H3K9me2 and H3K9me3 in centromere and pericentromeric regions using CUT&RUN.

*With CENPB, we no change in the signal area, but a decrease in eccentricity (circularity) when PBRM1 is deficient. In addition, we find a modest increase in NDC80 signal area and eccentricity (now added **in revised Supplementary Fig. 5**). Moreover, we find a tendency towards increased distance between NDC80 signals in metaphase cells in the absence of PBRM1 (**revised Supplementary Fig. 5**). Finally, as outlined below, we find a substantial change in histone H3K9 methylation patterns*

when PBRM1 is absent in both centromere and pericentromeric locations (now added in revised Fig. 5). Together, these data suggest that PBRM1 loss impacts the structure of pericentric regions and creates a structure that compromises the integrity of centromeric DNA.

3. While CENP-A level and distribution did not change, the alpha satellite DNA area increase needs explanation. Fig. 2a-2b and Fig. S4a -S4b need statistical tests to make a comparative statement. Additionally, the merged panels in Fig. 2C do not match the panels on the left.

There are several possible explanations behind the increased signal area in the KO cells using probes against alpha satellite sequences. First, the alpha satellite-containing chromatin regions could be failing to assemble into tightly organised structures when PBRM1 is absent, leading to expanded signal area. The absence of changes in CENP-A patterns could imply that the FISH signal changes represent regions outside of CENP-A occupied HORs. This is consistent with the modest changes in NDC80 and CENP-B IF signal shape, and interestingly, in this regard, we also found that the distance between NDC80 signals in metaphase cells is greater in the absence of PBRM1 (revised Supplementary Fig. 5). Together, these data are consistent with less tightly organised chromatin structure in pericentric heterochromatin when PBRM1 is deficient. However, we note that (unlike the mapping data and the functional readout of centromere fragility) the alterations in PBRM1 KO cells that are detected with imaging approaches are very subtle, and it is therefore possible that there are also changes in CENP-A containing chromatin structure that are below the detection limits of these assays.

Second, the increased centromere-associated recombination that occurs in the absence of PBRM1 (revised Figure 8) could lead to amplification of repetitive sequences. While both possibilities are consistent with the altered FISH signal, we don't think this latter possibility is playing a major role, since one prediction from this model is increased heterogeneity of signal, which is not apparent in our data. However, it is still possible (and indeed might become more important as the cells evolve), and we added text addressing these possibilities to the revised discussion.

In addition, we added more replicates of the FISH analysis and information on statistical significance. Finally, the merged panels of the KO cells in the original Fig 2c were swapped (thank you for noticing), and this has been corrected now.

4. It was inferred that PBRM1 regulates patterns of H3K9me3-enriched heterochromatic regions, as PBRM1-KO specific peaks of SMARCA4 are normally enriched at heterochromatic regions enriched in H3K9me3. A minor increase in overall H3K9me3 was also observed in the mutants, raising the question of whether the distribution of the increased H3K9me3 in PBRM1 KO cells is shifted and whether that explains the altered occupancy of SMARCA4. The chromatin occupancy of H3K9me3 in PBRM1 mutant cells should be determined to establish the relationship between PBRM1 and H3K9me3.

This was an excellent suggestion, thank you. As briefly described above, we have now used CUT&RUN to map genomic locations of H3K9me3 and H3K9me2 in the parental RPE1 and PBRM1 KO cells. We found that the distribution of both marks in centromeric and pericentric regions is altered when PBRM1 is deficient. These data are now included in revised Figure 5.

Briefly, when uniquely mapping reads are analysed (primarily present in ct arms), we find that there is a decrease in H3K9me2 peaks and an increase in H3K9me3 peaks in the PBRM1 KO compared with the parental cells (**revised Figure 5a**). The increase in H3K9me3 signal is apparent in regions previously occupied by H3K9me2 and SWI/SNF (**revised Figure 5b**). When repetitive DNA is analysed (using k-mer analysis), we find a redistribution of both H3K9me2 and H3K9me3 away from repetitive regions of ct arms and into HORs (**revised Fig. 5c-e**).

Together with the changes evident in the imaging assays, these data show that PBRM1 helps establish chromatin patterns in centromeres and pericentromeres.

5. Based on the GSEA results, comparing the frequency of apoptotic cells in PBRM1 mutant cell lines to their parental lines will also be helpful.

*We tested this possibility and found that, while misregulation of components of the apoptosis pathway was identified in the GSEA analysis, there was no apparent difference in the frequency of apoptotic cells in the PBRM1 mutant cells compared with parental cells. We have added these data in the revised version in **Supplementary Fig. 2c**.*

6. While the authors have concluded that PBRM1-KO cancers rely on the activity of spindle assembly checkpoint for survival (lines 299-301), based on the tumor volume in Fig. 6J and S11J (statistics absent), both parental and PBRM1 KO B16-F10 cells seem to have responded similarly to SAC inhibition. A major tumor volume reduction is also apparent in vehicle-treated PBRM1 KO cells compared to vehicle-treated parental cells. Therefore, the observed reduction in tumor growth and subsequent survival seems to be due to the loss of PBRM1. Additionally, the same data were plotted in both Fig. 6i and Fig. S11g-l, which should be noted in the corresponding legends.

Thank you for pointing this out. It is true that the PBRM1 mutant tumours grow more slowly than the parental B16. We used an immunocompetent mouse model, and this result is consistent with previous reports suggesting that loss of PBRM1 leads to increased immunogenicity (e.g. PMID: 29301960), and this could certainly be influenced by the increased genome instability arising from centromere fragility.

*However, even when the slower growth is accounted for, the impact of Mps1 inhibitors is proportionately greater on the PBRM1 deficient tumours. We find that there is no statistically significant effect of Mps1 inhibitors on survival or tumour volume of the parental B16 cells, but there is a significant effect of Mps1 inhibitors on the PBRM1 KO B16 tumours. We have altered the text and modified **revised Figure 7i-n** to better clarify this. Because of these changes, we no longer plot the same data in both main and Supplementary figures.*

7. In lines 301-303, authors implied a clinical significance of SAC dependence of PBRM1 KO cancers, which is confusing, as it is clear PBRM1 KO is itself a vulnerability in tumor cells. If they meant PBRM1 mutated tumor, testing such cell lines with the MPS1 inhibitor might be necessary to suggest this clinical significance. *We have altered the text in the revised manuscript to clarify this point as outlined above.*

8. In lines 111-112, the statement on the correlation between the PBRM1 level and the level of other centromeric proteins needs to be supported by a correlation metric. Additionally, the groups of centromeric proteins could be clustered together to visualize the relationship better.

Thank you, a correlation metric has been added showing a positive correlation (revised Supplementary Fig. 3e).

Reviewer #2 (Remarks to the Author):

Centromeres are essential for chromosome transmission, but are highly susceptible to DNA breaks. The mechanisms that protect centromere integrity are not fully understood. Lane et al. report that (peri)centromere abnormalities are the common feature of cells deficient in PBRM1, a component of the PBAF chromatin remodelling complex, and propose that this is mediated by its role in organising (peri)centromere chromatin structure. The work is timely and has significant implications for understanding cancer associated with PBAF deficiency.

The work has been meticulously conducted in a variety of conditions where PBRM1 is defective using multiple approaches including proteomics, mapping and scoring of short reads on (peri)centromeres, microscopy, and survival of cell lines and mouse models with appropriate negative controls. It seems to me that the results consistently support their model and the sensitivity of PBRM1-deficient cells/animals to mitotic inhibitors is convincing. I congratulate the authors for obtaining the results from this rather complex study for a very plausible model. I enjoyed reading this manuscript.

We thank this reviewer for the positive comments and for their appreciation of the efforts that went into this project.

The only minor point that was unclear to me was the relationship between the reduced protein levels of centromere-associated factors and PBRM1 deficiency (Figure 1). This is unlikely to be related to altered gene expression (Figure 1I), indicating the alternative possibility of protein stability. Would the centromere assembly of these factors affect their protein stability? For example, CENPB protein levels appear to decrease (Figure 1K, Figure S3a), whereas CENPB binding to centromeres has been shown to be blocked by the CENPB-box methylation (PMID: 15634350). I wonder if PBRM1 is related to such DNA modification. A discussion on this aspect could be very helpful to complete this work.

This is a great point, thank you. Our preferred model is indeed that stability of these factors is impaired due to changes in the chromatin platform onto which they assemble. While we focused on histone modifications here, DNA methylation could certainly play a role (we plan to investigate this in future), and we have added discussion to this effect in the revised manuscript.

Reviewer #3 (Remarks to the Author):

I co-reviewed this manuscript with one of the reviewers who provided the listed reports. This is part of the Nature Communications initiative to facilitate training in

peer review and to provide appropriate recognition for Early Career Researchers who co-review manuscripts.

Thanks very much for contributing to the reviewing process.

Responses to Reviewers

6 January 2025

We thank the reviewers for their positive comments and recommendations.

REVIEWERS' COMMENTS

Reviewer #1 (Remarks to the Author):

This is an interesting story, and the authors have done a good job of addressing our concerns in the body of the manuscript.

Reviewer #2 (Remarks to the Author):

I'm fully satisfied with the revision and recommend the paper for publication in Nature Communication.

Reviewer #3 (Remarks to the Author):
